# In silico docking yields small molecule negative allosteric modulators targeting the core of Frizzled 7

Magdalena M. Scharf [1], Julia Kinsolving [1,6], Lukas Grätz[1,5,6], Jan Hendrik Voss [1], David Carrasco-Busturia [2], Björn Forsberg [3], Peter Kolb [4] & Gunnar Schulte [1] ✉

Targeting the Frizzled family ($FZD_{1-10}$) of WNT receptors pharmacologically has, despite substantial therapeutic potential, proven difficult. Given an almost complete lack of validated, effective small molecules targeting FZDs, no putative ligand binding site has so far been identified. In order to target $FZD_7$, a potential target for the treatment of intestinal tumors, we combine an approach of adapted docking setups and large molecular library docking screens, identifying compound C407. Applying pharmacological assays, genetically-encoded biosensors, site-directed mutagenesis, cryo-electron microscopy and molecular dynamics simulations, the compound binding site in the core of the seven transmembrane bundle is validated and C407 is confirmed as a negative allosteric modulator of WNT-induced and FZD-mediated WNT/$\beta$-catenin signaling. In summary, we provide here the proof-of-principle that targeting FZDs with small molecule compounds is possible and effective. Future hit optimization and functional validation in disease-relevant in vitro and in vivo models will pave the way towards clinical exploration.

The superfamily of G protein-coupled receptors (GPCRs), consisting of about 800 receptors in humans, comprises well-established targets for small molecule drugs and presents the cornerstone for drug therapy of human disease. Roughly 150 of all non-sensory GPCRs are targeted by about 36% of all approved drugs on the market[1]. While this opens up a huge potential for further drug discovery and addressing yet untargeted receptors, it comes with the challenges of identifying suitable target receptors as well as the development of receptor-targeting compounds. One of the more enigmatic classes of GPCRs is the class F, consisting of ten Frizzled paralogs ($FZD_{1-10}$) and Smoothened (SMO)[2]. FZDs bind and are activated by the Wingless/Int-1 (WNT) family of secreted lipoglycoproteins, while SMO mediates Hedgehog signaling, positioning these receptors at the core of embryonic development,

stem cell regulation, tissue homeostasis, regeneration, as well as tumor development[3]. Here we focus on $FZD_7$, a receptor that relays activation by WNTs into a network of downstream signaling along the Disheveled (DVL)-dependent WNT/$\beta$-catenin pathway as well as through DVL-independent pathways, such as heterotrimeric $G_s$ proteins[4–7]. Physiologically, $FZD_7$ is essential for the maintenance of the intestinal epithelium and exaggerated $FZD_7$ signaling is associated with the development of intestinal cancers, but also other forms of cancer[8,9]. Furthermore, $FZD_7$ is targeted by the bacterial toxin TcdB from *Clostridium difficile*, which disrupts the $FZD_7$-mediated maintenance of the intestinal epithelial barrier[10,11]. The involvement of $FZD_7$ in human pathology renders it an attractive yet difficult target for human therapy. Attempts have been made to target $FZD_7$ with biologics and small-

[1]Department of Physiology & Pharmacology, Sec Receptor Biology & Signaling, Biomedicum, Karolinska Institute, Stockholm, Sweden. [2]Division of Theoretical Chemistry and Biology, School of Engineering Sciences in Chemistry, Biotechnology and Health, KTH Royal Institute of Technology, Stockholm, Sweden. [3]Department of Physics, Chemistry and Biology, SciLifeLab, Linköping University, Linköping, Sweden. [4]Philipps-Universität Marburg, Institute of Pharmaceutical Chemistry, Marburg, Germany. [5]Present address: Molecular, Cellular and Pharmacobiology Section, Institute of Pharmaceutical Biology, University of Bonn, Bonn, Germany. [6]These authors contributed equally: Julia Kinsolving, Lukas Grätz. ✉e-mail: gunnar.schulte@ki.se

molecule compounds, but the usefulness of some of these compounds has been questioned[12–15]. The recent years have provided new and high-resolution structural insights into the inactive and transducer-bound, active conformations of $FZD_7$ as well as an improved understanding of receptor dynamics, molecular switches and potential mechanisms that could be addressed to reduce constitutive or agonist-induced activation of $FZD_7$[6,7,16–20].

In this work, we engage in a quest for finding $FZD_7$-targeting negative allosteric modulators (NAMs) using the overall structure-function information that is now available for $FZD_7$ and the enormous chemical space of drug-like compounds that can be explored. We employ a large-scale in silico docking campaign yielding hits that enter thorough validation by competition binding assays, luciferase-based TCF/LEF reporter gene assays (TOPFlash), suitable counter assays, genetically encoded $FZD_7$-DEP-Clamp sensors[21], molecular dynamics (MD) simulations, and cryogenic electron microscopy (cryo-EM). In this pipeline, we identify drug-like compounds that target $FZD_7$ deep in the receptor core, in contrast to the endogenous ligands of the WNT family, which bind to the extracellular cysteine-rich domain (CRD) of FZDs[22]. By exerting their effects from this distinct binding site in the receptor core, the small molecule ligands act as NAMs to prevent WNT-induced conformational dynamics at the FZD/DVL interface and WNT/$\beta$-catenin signaling. The predicted binding pose of the compound is verified by combining MD simulations, structure-guided mutagenesis, and cryo-EM. Thus, we develop here the first-in-class FZD-targeting NAM with an established binding site, confirmed mode of action, and pharmacological efficacy in cellular systems. Our drug discovery efforts provide a basis to overcome the unmet clinical need for modulators of FZD-mediated signaling, leveraging receptor core intervention as a suitable mode of action.

## Results

Recently, it was discovered that most previously published smallmolecule ligands of FZDs were based on artifacts and not on the true modulation of FZD-mediated signaling[14,15]. Furthermore, derivatization of SAG1.3, a weak partial agonist of FZDs, led to a series of FZD-targeting small molecules with an unconfirmed binding site and mechanism of action[23]. This highlights the increasing need to find true and pharmacologically active modulators of FZDs and their signaling as tool compounds and a basis for future drugs in, e.g., anticancer therapies. Although a potential small-molecule modulator could bind at different locations of the FZD structure, the core of the 7 $\alpha$-helices of the transmembrane domain (7TMD) presents a suitable small molecule binding region[24]. The aim of the present study was to use large-scale computational docking screens aimed at the discovery of small-molecule ligands of $FZD_7$ that modulate $FZD_7$-mediated signaling by binding to the 7TMD core, thereby acting as allosteric modulators of WNT-induced signaling.

### Docking screens to the 7TMD of $FZD_7$ revealed a small molecule ligand of $FZD_7$

Here, we wanted to specifically explore possibilities to target the 7TMD with small-molecule modulators. For this, we utilized docking calculations to a cryo-EM structure of the 7TMD of a G protein-bound, active conformation structure of $FZD_7$ (PDB ID 7EVW, now 8YY8[6]). However, this is an *apo* structure, hence containing no information on a potential small molecule binding site. Of note, there is also no further information on a small molecule binding site within the 7TMD of FZDs available[2].

To guide the docking calculations and ensure that the $FZD_7$ structure was also adapted to accommodate the binding of small molecules, docking setups using template ligands from SMO were created. This was based on the idea that a multitude of structures of SMO, a close relative from the same class F as FZDs, are available with ligands bound in different locations of the 7TMD, ensuring both

comparability and a broad coverage of potential small molecule binding locations. Four different structures of SMO with ligands in diverse positions of the 7TMD (SAG1.5 in 6XBL; cyclopamine in 4O9R; vismodegib from 5L7I; SANT-1 from 4N4W) were hence selected, aligned to the structure of $FZD_7$ and the ligand was copied to the $FZD_7$ structure in its respective position. Both ligand and surrounding side chains in $FZD_7$ were then energy minimized in conjunction, to allow the receptor to adopt a "binding site" conformation which allows a potential ligand to both fit into the binding site and form interactions with the receptor. This finally resulted in four different docking setups, two with a potential binding site close to the extracellular side of the receptor (sagFZD7/SAG1.5; cycloFZD7/cyclopamine; Supplementary Fig. 1A, B), one with a potential binding site in an intermediate location (visFZD7/ vismodegib; Supplementary Fig. 1C) and one with a potential binding site deep within the 7TMD (santFZD7/SANT-1; Supplementary Fig. 1D).

These four docking setups were targeted in large molecular library docking screens of the ZINC15 drug-like (>9 million molecules) and lead-like (>2.5 million molecules) libraries (Fig. 1A). After visual evaluation of the top 500 ranked molecule poses of each of the eight docking screens, 22 molecules were selected to be purchased and tested in cell-based assays.

As an initial measure of compound binding, a BRET-based ligand displacement assay with BODIPY-tagged cyclopamine as tracer ligand was employed. Displacement of the tracer ligand was tested at a single concentration (10 µM) for all compounds selected from the docking calculations, resulting in one hit molecule: compound C45 (Fig. 1B). Further validation revealed that C45 could displace the tracer ligand in a concentration-dependent manner, although with a low $pIC_{50}$ value of $4.58 \pm 0.09$ (Supplementary Figs. 2 and 3). Of note, this molecule stems from docking calculations to the potential binding site that is located the deepest in the 7TMD, i.e., docking setup santFZD7 (Supplementary Fig. 1D), and is hence predicted to bind quite deep within the 7TMD of $FZD_7$ (Supplementary Fig. 4).

### Hit optimization of C45 uncovers another $FZD_7$-targeting small molecule

To explore the potential of C45 as a ligand of $FZD_7$ further, a follow-up in silico screen based on this compound was conducted (Fig. 1C). For an initial exploration of such a follow-up screen, a locally accessible molecular library provided by the Chemical Biology Consortium Sweden (CBCS) was used. Molecules with an ECFP4 tanimoto similarity above 0.4 to C45 were docked to a docking setup based on the prepared receptor structure santFZD7 and using the docking pose of C45 as the template ligand pose (docking setup c45FZD7). After evaluation of the resulting docking poses, 11 molecules were selected to be tested in competition binding experiments (Fig. 1D). Among these compounds, compound C407 could be identified as an additional ligand of $FZD_7$, displacing the tracer ligand in a concentration-dependent manner and with a slightly improved $pIC_{50}$ compared to C45 ($pIC_{50}$(C407) = $4.86 \pm 0.04$ compared to $pIC_{50}$(C45) = $4.58 \pm 0.09$; Fig. 1E; Supplementary Fig. 3).

### Hit optimization of C407 uncovers more ligands of $FZD_7$ with a similar molecular core scaffold

To follow up further on these two hit molecules, C45 and C407, as well as their shared core molecular scaffold (Supplementary Fig. 3), another round of hit optimizations was conducted. This time, the Enamine screening compound catalog was used to find purchasable molecules with a high similarity to C407 (Fig. 1F). The resulting focused compound library of 177 compounds was then docked to the docking setup c45FZD7, and eight molecules were selected for testing after evaluation of the docking results (see Supplementary Data 1 for docking poses).

The selected compounds were tested in the BRET-based ligand displacement assay, revealing several additional hit molecules that

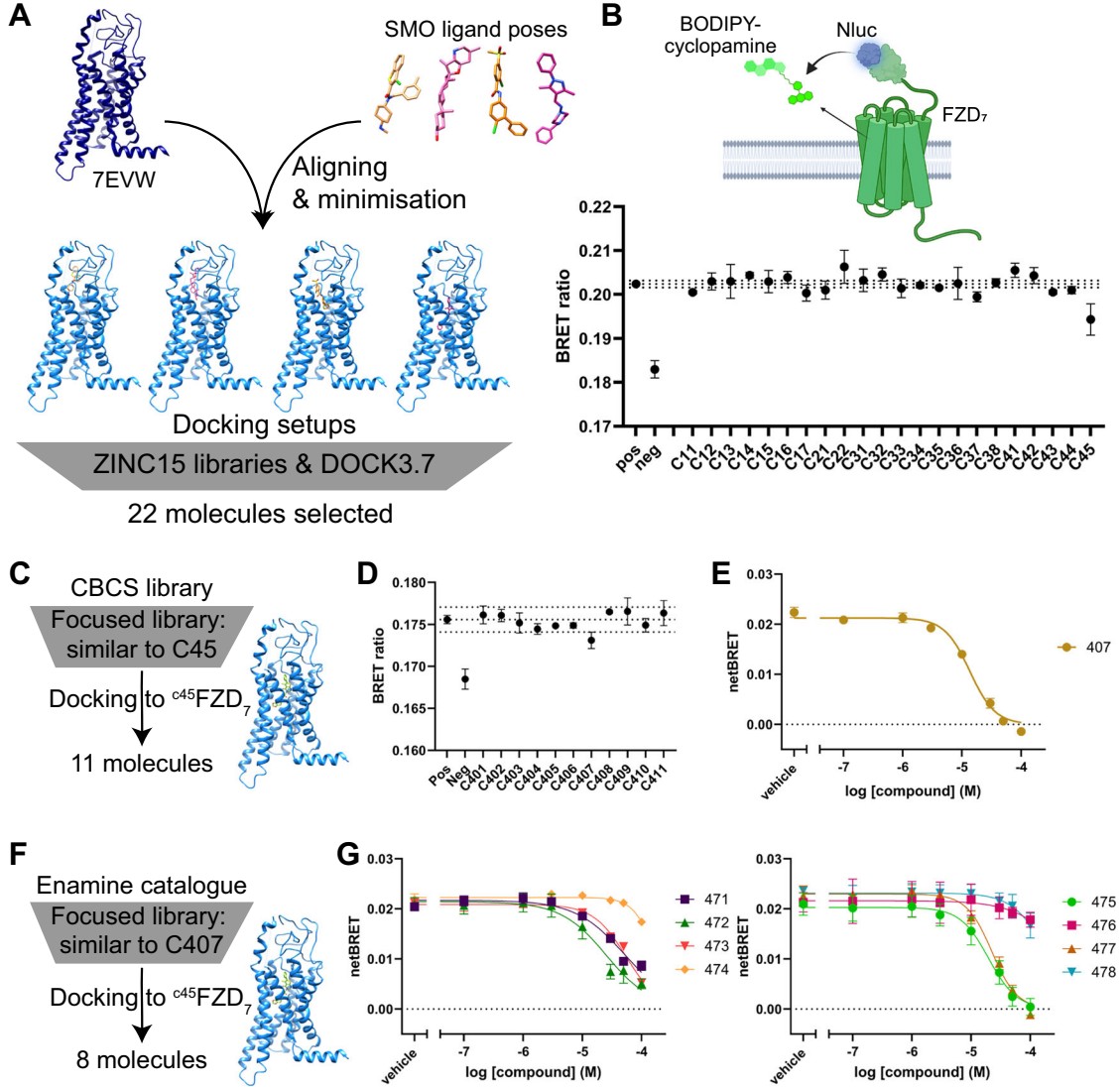

**Fig. 1 | Initial compound screens. A** Scheme summarizing the preparation of docking setups and the large library screen. **B** Representative measurement of all compounds selected from the primary screen in the BRET-based competition binding assay (Nluc-FZD$_7$ and BODIPY-cyclopamine) at a single concentration of 10 μM. The scheme above the graph illustrates the assay principle. A full concentration response curve for the binding of the potential hit compound C45 can be found in Supplementary Fig. 2. **C** Scheme of the in silico follow-up screen based on initial hit C45 using the CBCS library. **D** Representative measurement of all compounds selected from the screen in (**C**) in a BRET-based competition binding assay (HiBiT-FZD$_7$ and BODIPY-cyclopamine) at a single concentration of 10 μM. **E** Concentration-response curve for C407 from a BRET-based competition binding assay (Nluc-FZD$_7$ and BODIPY-cyclopamine). **F** Scheme of the follow-up screen based on hit molecule C407 and using the Enamine catalog. **G** Concentration-response curves for the compounds selected in (**F**) from BRET-based competition binding assays (Nluc-FZD$_7$ and BODIPY-cyclopamine). **B**, **D** Data points are mean ± SD of technical triplicates from a representative experiment of $n$ = 3 independent experiments. Dotted lines represent mean ± 3 · SD of the positive control as a measure of significance. Pos = positive control: DMSO and BODIPY-cyclopamine; neg = negative control: buffer only (with DMSO; no BODIPY-cyclopamine). **E**, **G** Data points represent mean ± SEM of $n$ = 3–5 independent experiments. **B** Scheme created in BioRender. Kinsolving, J. (2025), https://BioRender.com/8drrhgy. Source data are provided as a Source Data file.

were able to displace the tracer ligand in a concentration-dependent manner, namely compounds C471, C472, C473, C475 and C477 (Fig. 1G and Supplementary Fig. 3). However, none of the tested compounds showed better pIC$_{50}$ values than C45 or C407, hence, no improvement was achieved (Supplementary Fig. 3). All of the additionally tested molecules as well as the initial hit molecules C45 and C407 contain a similar structural core motif (5-methyl-3$H$-thieno[2,3-d]pyrimidin-4-one; Supplementary Fig. 3). While it is difficult to derive a structure-activity-relationship due to the small set of compounds and the small differences in pIC$_{50}$ values, a few trends can be observed. Specifically, compounds that contain an ethyl-ester moiety as well as an amide connected to the core scaffold seem to have overall better pIC$_{50}$ values compared to compounds that have only one of these moieties attached to the core.

## Compounds act as FZD$_7$-targeting negative allosteric modulators of WNT-induced signaling

To evaluate whether the hit molecules exert any pharmacological effect on FZD$_7$-mediated signaling beyond binding to the receptor, a TOPFlash assay was employed. In brief, this assay uses a T-cell factor/lymphoid enhancer factor (TCF/LEF)-driven reporter gene (Firefly luciferase, Fluc) readout to assess β-catenin-mediated signaling. The effect of compound addition on WNT-3A-induced signaling was assessed, revealing a reduction of the TOPFlash response upon addition of several of the compounds (Fig. 2A). None of the compounds induced a TOPFlash response in absence of WNT (Supplementary Fig. 5A). To ensure that the reduction of the TOPFlash signal was not caused by interference with Fluc luminescence readout, control experiments were conducted (Supplementary Fig. 5B). Furthermore,

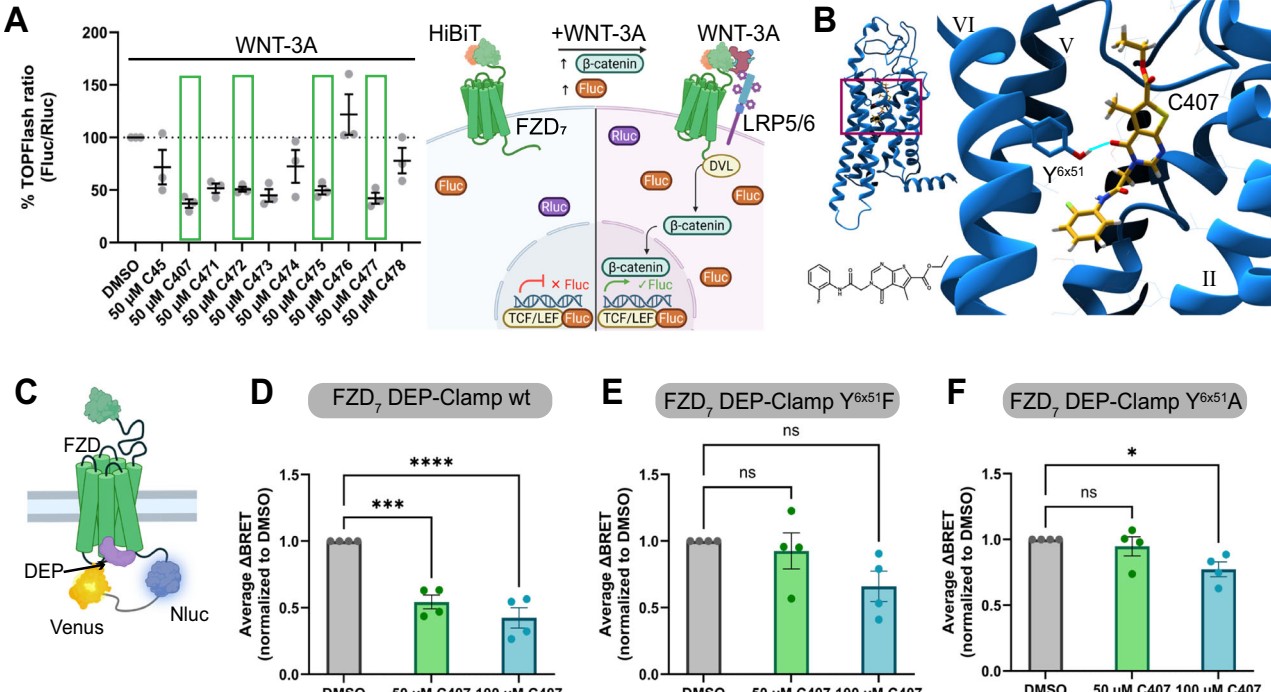

**Fig. 2 | Pharmacological characterization of compounds. A** Impact of compounds on WNT-3A-induced (300 ng/mL) TOPFlash at HiBiT-FZD7 normalized to WNT-3A response in absence of test compound (DMSO). A scheme illustrating the assay principle is shown on the right. To determine true hits (C407, C472, C475, and C477; green boxes), control assays (Supplementary Fig. 5) and cell toxicity (Supplementary Fig. 6) have to be taken into account. Data (lines) represent mean ± SEM of three independent experiments. **B** Predicted binding pose of C407 (goldenrod) to FZD7 (blue) from the docking calculations. The compound is predicted to bind deep within the 7TMD (inset: side-view of the FZD7-C407 complex) and forms polar interactions to residue Y489[6x51] (bright blue line). The chemical structure of C407 is given for reference. **C** Scheme of the FZD-DEP-Clamp sensor. Upon WNT-stimulation a rearrangement of the complex results in a change in BRET. D: FZD7-DEP-Clamp measurements determining the WNT-3A-induced (1 μg/mL) sensor rearrangement in the absence (DMSO) and presence of different concentrations of C407. **E, F** Same measurements as in (**D**), but after introducing Y489[6x51]F (**E**) or Y489[6x51]A (**F**) point mutations into the FZD7-DEP-Clamp. **D–F** Data (lines) represent mean ± SEM of four independent experiments. Time-courses can be found in Supplementary Fig. 7. Statistical analysis was performed with a one-way analysis of variance (ANOVA) followed by Dunnett's post-hoc test comparing all other means with the means of the DMSO control. *$p = 0.026$, ***$p = 0.0004$, ****$p < 0.0001$. **A** Scheme created in BioRender. Kinsolving, J. (2025) https://BioRender.com/8drrhgy. **C** Scheme created in BioRender. Grätz, L. (2025) https://BioRender.com/mvf137p. Source data are provided as a Source Data file.

selected compounds were tested for cell toxicity to ensure that observed effects were not caused by unspecific compound toxicity (Supplementary Fig. 6). In this assay, C45 showed high cell toxicity and was hence not used in further experiments. However, other compounds—including C407—did not affect the cell viability of HEK293 cells. Based on these combined assay results, compounds C407, C472, C475, and C477 could be confirmed as negative modulators of WNT-3A-induced β-catenin signaling. As the most potent of the hit compounds, all further experiments focused on C407.

Since the TOPFlash assay is a strongly amplified downstream readout in the signaling pathway, it had to be ensured that the observed effects were indeed occurring at the level of FZD7 and not downstream of the receptor. Therefore, we employed FZD-independent induction of the TOPflash signal achieved by over-expression of DVL in ΔFZD1–10 cells. Indeed, C407 did not affect the DVL-induced TOPFlash response (Supplementary Fig. 5C), indicating that the inhibition of the WNT-3A-induced TOPFlash response by C407 occurs via FZD7.

**C407 effect is mediated via a binding site in the FZD 7TMD core**
To confirm the potential binding site of C407 and the interactions with FZD7, further pharmacological characterization was conducted. In the docking-predicted binding pose, C407 binds deep within the 7TMD of FZD7, similar to C45 (Fig. 2B, Supplementary Fig. 4 and Supplementary Data 1). As C45, C407 is also predicted to form polar interactions to Y489[6x51] (Fig. 2B, Supplementary Fig. 4B, C; numbering in superscript

according to GPCRdb nomenclature[25]). Further stabilization of the compound pose is likely mediated by apolar and aromatic interactions.

To confirm the direct action of C407 on the mediated signaling pathways via FZD7, a conformational BRET-based sensor measuring direct effects mediated via FZDs was used, the FZD-DEP-Clamp sensor[21,26]. In brief, FZD-Nluc is fused with a Venus-tagged version of the main FZD-interacting domain of DVL, Venus-DEP, via a 10 nm linker (Fig. 2C). The globular DEP domain couples constitutively to FZD, resulting in a high basal BRET response. Upon stimulation with WNT, the complex rearranges conformationally, mirroring receptor activation and leading to a change in BRET. An effect of the compound on the WNT-induced BRET-change would, hence, show that (1) the compound acts directly via FZDs and (2) confirm the assumption that the compound acts as a NAM, providing a mode of action for the observed reduction in the TOPFlash assay. Indeed, addition of C407 impacts the WNT-3A-induced rearrangement in the FZD7-DEP-Clamp, with a more pronounced effect observed for a higher concentration of C407 (Fig. 2D and Supplementary Fig. 7A) and without showing an effect on the BRET signal when used on its own (Supplementary Fig. 7B). This confirms the effect of C407 via FZD7 in the corresponding TOPFlash signaling readout. To further corroborate that C407 not only acts via FZD7 but also through the binding site predicted by the docking calculations, site-directed mutagenesis was used. In accordance with the predicted binding pose, residue Y489[6x51] was mutated to phenylalanine (Y489[6x51]F) and to alanine (Y489[6x51]A) to disrupt the polar interaction. The effect of the respective mutation was evaluated using the FZD7-

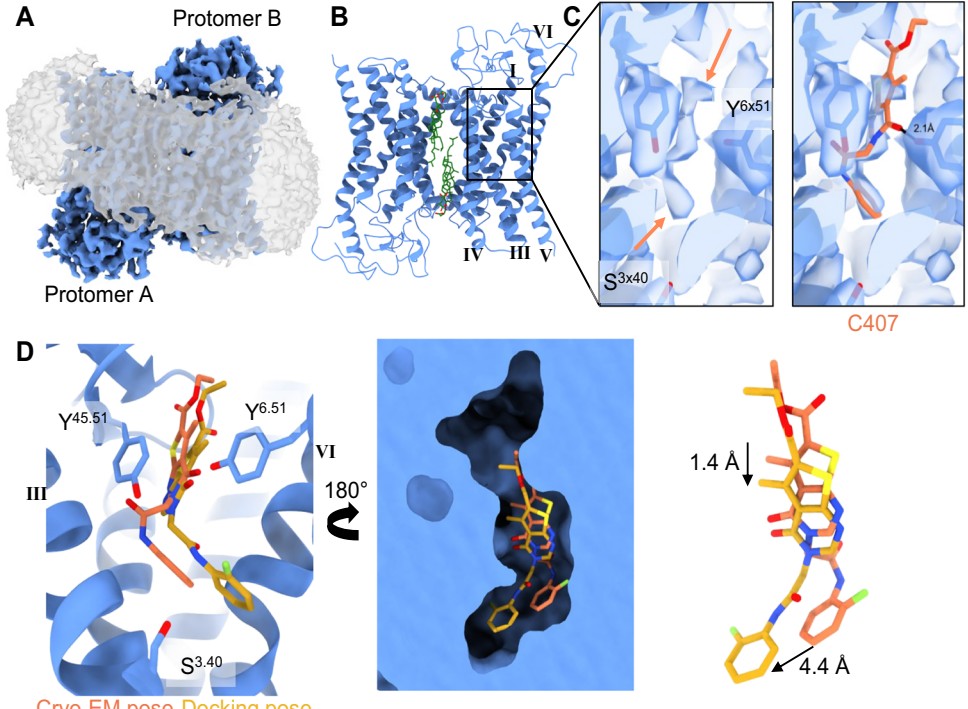

**Fig. 3 | Cryo-EM structure of the FZD$_7$-dimer containing a density potentially corresponding to C407. A** Cryo-EM map of the antiparallel FZD$_7$-dimer (blue) solubilized in LMNG detergent (micelle shown in gray). **B** Model of the FZD$_7$-dimer after focused masking. The lipidic interface consisting of conserved cholesterol binding sites (green) mediates interactions between the protomers at the dimer interface. **C** Reconstruction and zoom in on protomer B showing the tentative placement of C407 in a weak density observed in the 7TMD cavity. The density

(highlighted with orange arrows; left) is consistent with low-occupancy ligand binding, but does not allow unambiguous assignment. The predicted binding pose of C407 is modeled in this density (orange molecule; right), **D** Model (left) and surface representation (middle) of protomer B of the FZD$_7$-dimer depicting the cavity within the 7TMD and comparing the cryo-EM pose (orange) and docking pose (yellow) of C407 that can be accommodated in the pocket. Docking pose and cryo-EM pose are slightly shifted (right).

DEP-Clamp sensor in connection with agonist stimulation. Indeed, either mutation abrogated the effect of C407 on the WNT-3A-induced BRET-change (Fig. 2E, F and Supplementary Fig. 7). These results corroborate that C407 binds to the site in the FZD$_7$ 7TMD that was predicted by the docking calculations.

Furthermore, the FZD-DEP-Clamp sensor was employed to evaluate the paralog selectivity of C407. For this, we selected representative members of each of the homology clusters of FZD paralogs that are involved in WNT/$\beta$-catenin signaling, in specific FZD$_2$ (FZD$_{1,2,7}$ cluster), FZD$_4$ (FZD$_{4,9,10}$ cluster), and FZD$_5$ (FZD$_{5,8}$ cluster)[2]. For all tested paralogs, C407 exhibited a similar effect on WNT-3A-induced BRET changes in the respective DEP-Clamp sensor compared to the FZD$_7$-DEP-Clamp (Supplementary Fig. 8), indicating no selectivity between those paralogs. This is in line with the high degree of evolutionary conservation of Y$^{6x51}$ among FZDs as well as the high overall conservation of residues within the potential binding site (see Supplementary Fig. 9 and Supplementary Table 1)[7].

### Structural elucidation of putative C407 interactions

In order to obtain more detailed information about and to verify the binding pose of C407 in FZD$_7$, we set out to solve the cryo-EM structure of FZD$_7$ in complex with C407. Based on the previously published, inactive FZD$_7$ apo structure[16], we employed a similar purification protocol in the presence of 10 µM C407 during the solubilization step, and 1 µM C407 throughout all other purification steps. A 10:1 molar ratio of C407:receptor was supplemented to the sample prior to grid freezing (Supplementary Fig. 10A).

The structure of the inactive and antiparallel FZD$_7$ dimer was determined by single particle cryo-EM, resulting in a reconstruction with a global resolution of 2.5 Å (Fourier shell correlation = 0.143)

(Fig. 3A). The stable lipid interface between the antiparallel protomers includes the conserved functional cholesterol site originating from cholesterol hemisuccinate and is unambiguously modeled, matching previous findings and stabilizing the dimer in an inactive conformation (Fig. 3B)[16]. Nearby lipid densities were also found to be consistent with earlier reported palmitic acid molecules, but were not modeled. In contrast to the previously published apo FZD$_7$ dimer (PDB ID 9EPO), which was solved at 1.9 Å with C2 symmetry, the reconstruction with C407 added in the preparation showed limited resolution due to differences observed between the two monomers. Subsequently, C1 symmetry was used throughout refinement with an overall root-mean-square deviation (RMSD) value of 0.8 Å from the inactive apo FZD$_7$ dimer structure.

To remove noise generated from the micelle and to characterize differences between the two monomers, an initial loose mask was generated on the 7TMD cavity corresponding to protomer B and used for 3D classification. Regardless of multiple rounds of classification, no improvement in the density was observed (Supplementary Fig. 11). Consequently, a loose focused mask was placed on the entirety of the dimer and used for subsequent local refinements. During map modification, local refinements were used to improve map quality, however, it should be noted that previous findings have highlighted the pitfalls of machine learning (ML)-based volume improvements that can enhance densities for biomolecules but prove detrimental for ligands or small molecule densities[27]. Therefore, the use of ML-based methods to improve the map was avoided. Protomer B shows an overall better reconstruction and model fit compared to protomer A, notably due to a better-defined density of the intracellular tip of TM6 (Supplementary Fig. 10B). Due to the symmetry-breaking features, we continued with an analysis of protomer B to use for 3D classification (without

alignment) in CryoSPARC. The resulting 3D classification showed two discrete heterogeneous classes with reported overall resolutions of 3.2 Å (class 1) and 2.5 Å (class 2) (Supplementary Fig. 10B). These two distinct particle classes also revealed a significant difference within the 7TMD of the receptor: while the first reconstruction lacked any evidence of a density in the cavity (class 1), the second reconstruction shows supports for a non-proteinaceous density within the 7TMD of the receptor (class 2).

Regardless of masking and subsequent local refinements, we could not confidently model C407 in this non-proteinaceous density within the 7TMD of $FZD_7$ in the receptor-focused reconstruction at 2.5 Å ; however, the density aligns well with the proposed docking site for C407 (Fig. 3C). While this space also overlaps with the internal water cavity described in the *apo*-$FZD_7$ structure, the resolution limits modeling of water molecules and the elongated density is distinctly different from what is observed in the *apo*-$FZD_7$ structure. This suggests the presence of a small molecule rather than the presence of water. Positioning C407 into the density represents the most plausible conformation, given the reconstruction was fit independently of the docking pose. This modeled pose overall resembles the pose suggested by the docking calculations (Fig. 3D), including a similar polar interaction to $Y489^{6x51}$, which was also confirmed in functional assays (Fig. 2). Furthermore, the reconstruction of $FZD_7$ with a density that could match C407 based on docking and pharmacological data reveals here potentially the first published small-molecule-bound FZD structure. Cryo-EM refinement and validation statistics are shown in Supplementary Fig. 12 and Supplementary Table 2, while map and model can be found in Supplementary Data 2 and 3, respectively.

### MD simulations reveal a rearrangement of C407 in the binding site

To evaluate the stability of the binding pose of C407 to $FZD_7$ as predicted by the docking calculations and as modeled based on the cryo-EM data further, unbiased MD simulations of 300 ns length per replica were performed for each of the two complexes. While in both cases the overall conformation of the protein stays similar to the starting model (Supplementary Fig. 13), the binding pose of C407 rearranges, resulting in a new and stable binding pose that deviates from the initially modeled poses (Supplementary Figs. 14A and 15) by 4–6 and 3–6 Å RMSD, respectively. In MD simulations starting from either (i.e., docking-predicted or cryo-EM-modeled) of the initial models, this change in binding pose can be observed as a movement of 4–6 Å of C407 deeper into the 7TMD core of $FZD_7$ compared to the initial pose (Fig. 4A, B and Supplementary Fig. 16). For the MD simulations starting from the docking pose of C407, this reorientation happens as a sudden movement within the first 100 ns of the trajectory, as can be deduced from distance measurements between atoms of C407 and the receptor (Fig. 4C and Supplementary Fig. 14). After this reorientation, the binding pose of C407 remains stable and its binding location does not change further. Furthermore, the changed binding pose seems to be reproducible overall in all three independent replicas. In contrast, the binding pose assumed by C407 when starting from the binding pose modeled based on the cryo-EM data seems less reproducible between the three replica (Fig. 4C and Supplementary Fig. 14). However, a general movement of the compound deeper into the 7TMD of $FZD_7$ is observed for all three replicas, although resulting in a seemingly less stable and reproducible final binding pose compared to the MD simulations starting from the docking pose. Finally, the most frequent binding poses that C407 adopts in both MD simulations differ slightly, although there are similarities observable between a pose less frequently assumed in the MD simulations starting from the docking pose and the orientation the compound assumes in the MD simulations starting from the cryo-EM pose (Fig. 4A, B). Overall, although there are differences observable in the MD simulations starting from both initial poses, the general trend of C407 moving to a binding location deeper

within the 7TMD of $FZD_7$ is comparable and reproducible in both cases.

### C407 binding poses uncovered by MD simulations can be confirmed pharmacologically

To examine the binding poses assumed by C407 throughout the MD simulations further, polar interaction frequencies between the compound and $FZD_7$ were analysed for the course of each trajectory (Fig. 5A, B). Starting from either of the initial binding poses, C407 assumes the most frequent polar interactions with residues $H303^{2x58}$ and $S351^{3x40}$. Interactions of C407 are highly frequent and reproducible in the MD simulations starting from the docking pose, while they seem to be more pronounced in one of the replicas compared to the other two when starting from the cryo-EM pose. Interestingly, this frequency in interactions seems to be linked to the orientation of the amide in C407. When starting from the cryo-EM pose, highly frequent interactions with the two mentioned residues can only be observed for the replica in which the amide flips from the initial *cis* conformation to a *trans* orientation (Supplementary Fig. 17). In the docking pose, this amide is already initially modeled in a *trans* conformation. Hence, a *trans* orientation of the amide might be required to form more stable polar interactions and to arrive to a potentially more stable binding pose (compare also replica 1 vs. replica 2 and 3 for the MD simulations starting from the cryo-EM pose in Fig. 4C and Supplementary Fig. 14).

Of the most frequent polar interactions between C407 and the receptor, especially the interaction to $S351^{3x40}$ is of interest for further investigations, since it can only be formed if C407 indeed moves deeper into the core of the receptor as suggested by the MD data. Hence, the effect of a $S351^{3x40}A$ mutation on C407 effect was tested using the $FZD_7$-DEP-Clamp sensor. Indeed, when mutating this residue, C407 did not exert any effect on the WNT-induced sensor-rearrangement anymore (Fig. 5C and Supplementary Fig. 7). This result corroborates the observations of a binding location of C407 deep within the 7TMD of $FZD_7$ as suggested by the MD simulations.

### MD simulations reveal a potential mechanism of action of C407

During the analysis of the binding pose change of C407 throughout the MD trajectories, an additional observation was made. In the MD simulations starting from the docking pose, the compound inserts its fluorophenyl moiety between TM3 and TM6 of the receptor while moving down (Supplementary Fig. 18A). However, to be able to do so, a conformational rearrangement of two residues has to occur (Fig. 5D top and middle). These two residues are $W354^{3x43}$ and $Y478^{6x40}$, which both are part of an extended molecular switch network with important roles in FZD activation[16,18]. While $W354^{3x43}$ and $Y478^{6x40}$ are involved in an aromatic-aromatic-stacking interaction in the initial model, this interaction breaks due to a reorientation of the side chains throughout the trajectories (Fig. 5D–F). Importantly, this reorientation can only be observed in presence of C407 but not for the *apo* receptor starting from the same $FZD_7$ conformation (Fig. 5E, F and Supplementary Figs. 18, 19, and 20). While in two of the replicas, this side chain reorientation seems to be limited to $Y478^{6x40}$, the observed behavior in one replica is slightly different. In this replica ("replica 2"), C407 initially adopts a more curved binding pose, avoiding insertion of the fluorophenyl moiety between TM3 and TM6 (see also Fig. 4A cluster 1 for the respective compound pose). However, the compound then reorients throughout the last frames of the trajectory, inserting its fluorophenyl moiety between TM3 and TM6 as observed for the other replica (Supplementary Fig. 18A) and linked to conformational rearrangements of the side chains of both $W354^{3x43}$ and $Y478^{6x40}$ (Supplementary Figs. 19 and 20). Generally, the observed conformational rearrangement of these microswitch residues $W354^{3x43}$ and $Y478^{6x40}$ in the presence of C407 suggests a distinct mechanism of action for the FZD-targeting small molecule compound that might explain the

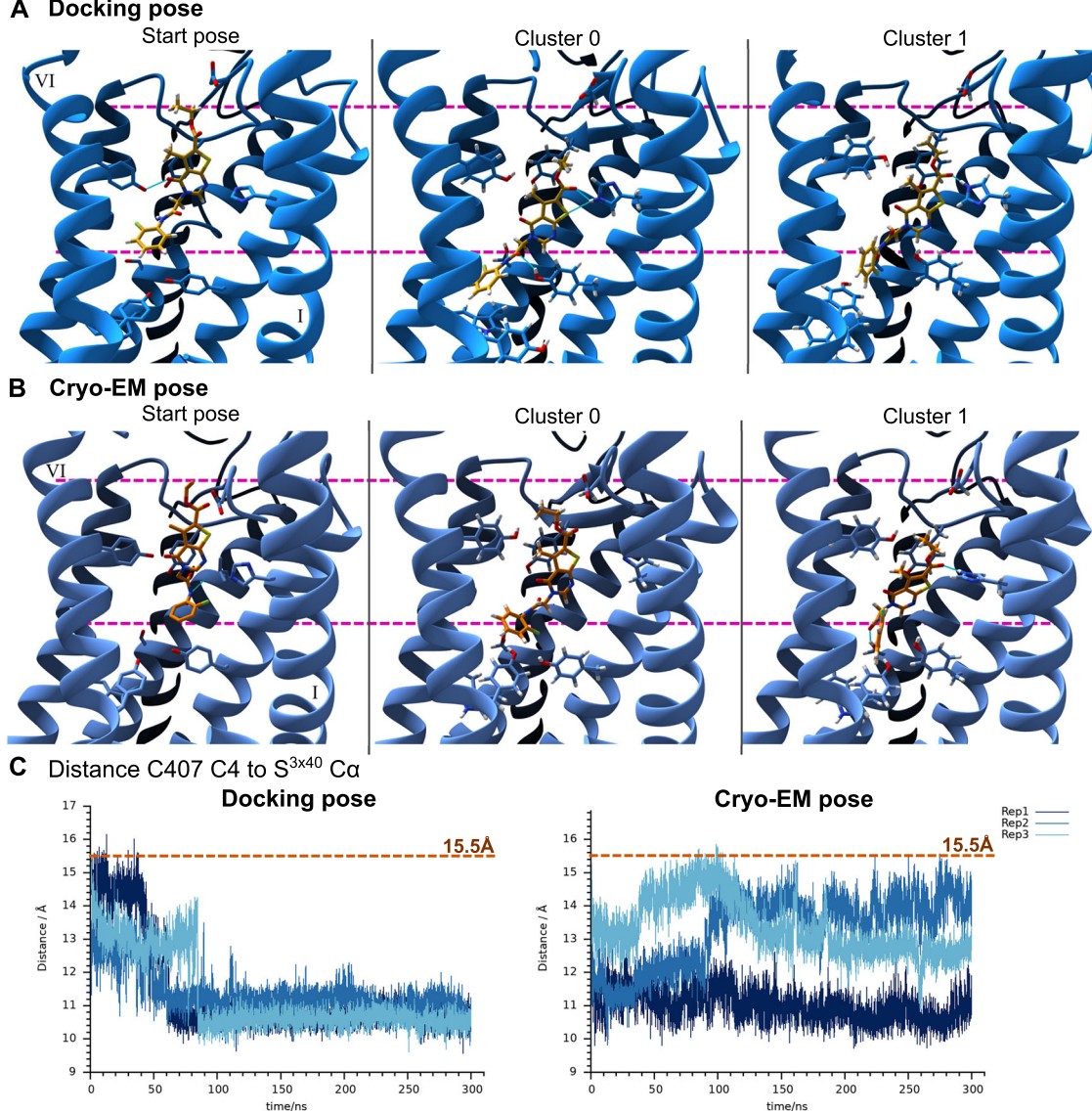

**Fig. 4 | Movement of C407 throughout the MD trajectories to a binding pose located deeper within the 7TMD core. A** MD simulations starting from the docking pose of C407 (yellow sticks). Left: Start pose as predicted by the docking calculations. Middle and right image represent the most frequent C407 binding poses throughout the trajectory with 51% of frames (cluster 0) and 35% of frames (cluster 1) (Supplementary Data 4). Pink dashed lines represent the highest and lowest reaching parts of C407 in the starting pose. TM7 is hidden for improved clarity and numbering of the TMs is indicated in roman numerals. Full side-views of the receptor can be found in Supplementary Fig. 16. **B** Same as (**A**), but for the MD simulations starting from the pose of C407 (orange sticks) as modeled based on the structural cryo-EM data. The most frequent binding poses of C407 throughout the trajectory were found in 56% (cluster 0) and 30% (cluster 1) of the frames (Supplemenatry Data 4). **C** Distance between atom C4 of C407 and the S351$^{3x40}$ C$\alpha$ atom of FZD$_7$ for the MD simulations starting from the docking pose (left) or the cryo-EM pose (right). The initial distance as measured in the starting model is indicated by the orange dashed line. Refer to Supplementary Fig. 16 for location of residue and atoms used for distance measurements and to Supplementary Fig. 14 for complementary distance measurements. Data for the three independent replica are shown in shades of blue.

compound's effects on FZD$_7$-mediated signaling as observed in cell-based assays.

Comparing these results to the MD simulations when starting from the cryo-EM pose, a different behavior can be observed. In this case, the fluorophenyl moiety does not insert between TM3 and TM6 (Fig. 5D bottom and Supplementary Fig. 18A). Instead, C407 adopts a more curved conformation, pointing the fluorophenyl moiety towards a space between TM6 and TM7 of FZD$_7$ (similar to the first orientation adopted by C407 in "replica 2" when starting from the docking pose). This could be linked to a very stable conformation of the two micro-switch residues W354$^{3x43}$ and Y478$^{6x40}$, which show barely any movement compared to the initial orientation (Supplementary Figs. 18, 19, and 20). This property can likely be attributed to the fact that FZD$_7$ is in

an inactive conformation in this case. Comparison of the results of an MD simulation of an *apo* inactive conformation as performed by Bous et al.[16] reveals similar little movement of these residues (Supplementary Fig. 21), while more movement can be observed in the case of the *apo* active conformation (Supplementary Figs. 18, 19, and 20). The rigidity of these two residues in inactive conformations of FZD$_7$ compared to active conformations is also supported by the lower values of the root-mean-square fluctuation (RMSF) of the side-chain heavy atoms of both residues (Supplementary Table 4). Hence, the divergent behavior observed in the two MD simulations starting from the docking pose or the cryo-EM pose might be linked to the underlying receptor conformation and does, therefore, not contradict the proposed mechanism of action of C407.

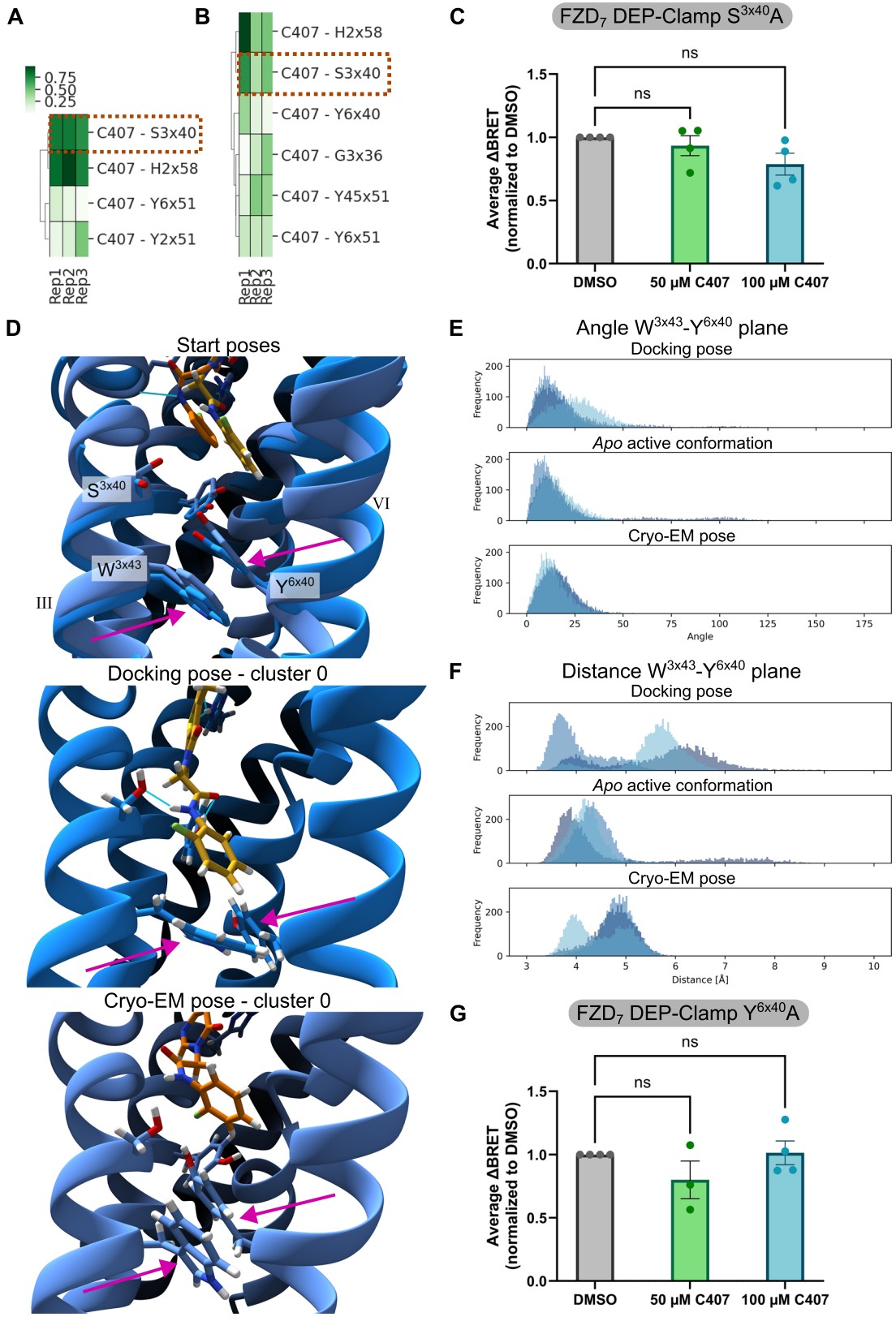

To confirm that the disruption of the conformation of the microswitch residues W354$^{3x43}$ and Y478$^{6x40}$ by C407 is not based on a simulation artifact, we additionally ran MD simulations with one of the compounds that was identified not to bind to FZD$_7$ or affect induced signaling, compound C476. Interestingly, within the first nanoseconds of the simulation, C476 shows a similar movement deeper into the 7TMD of FZD$_7$ as observed for C407, although it cannot form the same

polar interactions with S351$^{3x40}$ (Supplementary Fig. 22). This might indicate a generally more favorable binding location for the compounds deep within the 7TMD. However, C476 does not disrupt the conformation of the microswitch residues W354$^{3x43}$ and Y478$^{6x40}$ (Supplementary Fig. 23). On the contrary, the conformation of these residues seems to be more stabilized compared to the *apo* active simulations and more comparable to the inactive conformation MD

**Fig. 5 | Interactions and potential mechanism of action of C407 as derived from MD simulations.** Most frequent hydrogen bonds between C407 and $FZD_7$ throughout the MD simulations starting from the docking pose (**A**) or the cryo-EM pose (**B**). The interaction of C407 with $S351^{3x40}$ is highlighted with an orange box. **C** $FZD_7$-DEP-Clamp measurements determining the WNT-3A-induced (1 μg/mL) sensor rearrangement in absence (DMSO) and presence of different concentrations of C407 after introduction of point mutation $S351^{3x40}$A. **D** Closeup view of the microswitch consisting of residues $W354^{3x43}$ and $Y478^{6x40}$ in the starting models (top) and in reference to the most frequent poses of C407 throughout the MD simulations starting from the docking pose (middle; C407 in yellow) and the cryo-EM pose (bottom; C407 in orange). Side-chains of the two residues are highlighted with magenta arrows. TM5 is hidden in these images for improved clarity. Numbering of the TMs is indicated in roman numerals. **E** Frequency distributions of observed angles between the vectors perpendicular to the aromatic planes of $W354^{3x43}$ and $Y478^{6x40}$. **F** Frequency distributions of observed distances between the center of mass of the aromatic planes of $W354^{3x43}$ and $Y478^{6x40}$. **E, F** Distributions for the MD simulations starting from the docking pose (top) or the cryo-EM pose (bottom) are compared with those of the *apo* $FZD_7$ structure as used for the docking calculations (middle). Data are shown for three independent replicas in different shades of blue. Corresponding time-courses can be found in Supplementary Fig. 18. **G** $FZD_7$-DEP-Clamp measurements determining the WNT-3A-induced (1 μg/mL) sensor rearrangement in absence (DMSO) and presence of different concentrations of C407 after introduction of point mutation $Y478^{6x40}$A. **C, G** Data (lines) represent mean ± SEM of four independent experiments. Statistical analysis was performed with a one-way analysis of variance (ANOVA) followed by Dunnett's post-hoc test comparing all other means with the means of the DMSO control. Corresponding time-course measurements can be found in Supplementary Fig. 7. Source data are provided as a Source Data file.

simulations, as can also be seen from the RMSF values (Supplementary Table 4). These results point further towards the proposed mechanism of action of C407.

To validate this proposed mechanism of action experimentally, the $FZD_7$-DEP-Clamp sensor was employed once more, this time inserting point mutations $W354^{3x43}$A or $Y478^{6x40}$A. In the sensor containing the $W354^{3x43}$A mutation, the basal interaction between $FZD_7$ and DEP was seemingly lost, which did, hence, not allow any further characterization (Supplementary Fig. 7G). However, the sensor containing the $Y478^{6x40}$A mutation was still functioning, allowing to assess the effect of C407 on the WNT-3A-induced sensor rearrangement. Indeed, the effect of C407 was lost after introducing mutation $Y478^{6x40}$A (Fig. 5G and Supplementary Fig. 7F). This result is in line with a potential mechanism of action, where the compound affects the conformation of switch residues $W354^{3x43}$A and $Y478^{6x40}$A, thereby acting as a FZD-targeting NAM.

## Discussion

Here, we describe the discovery and characterization of a small molecule modulator of WNT-induced signaling that binds to the 7TMD of FZDs. This was achieved by integrating a variety of techniques, including docking calculations, pharmacological assays and genetically modified biosensors, site-directed mutagenesis, cryo-EM and MD simulations. By targeting an experimental structure of the 7TMD of $FZD_7$ with an approach of adapted docking setups and large molecular library docking screens, the $FZD_7$-targeting ligand C407 and several derivatives were identified. Furthermore, C407 could be characterized as a NAM of WNT-induced signaling and its predicted binding site within the 7TMD of $FZD_7$ was confirmed using site-directed mutagenesis.

The ability of a small molecule targeting the 7TMD core of FZDs to affect WNT-induced TOPFlash signaling clearly underlines that protein dynamics in FZDs are required for agonist-driven initiation of the WNT/β-catenin pathway, which is fully in line with basic concepts of GPCR pharmacology[28]. Thus, the identification of a FZD-targeting NAM reducing WNT-induced β-catenin signaling underlines that the dogma of signalosome-based signal initiation independent of GPCR-like receptor dynamics needs to be revised and refined to fully understand signal initiation—in agreement with what we have previously suggested[2,23,29].

Despite the discovery of a first-in-class FZD-core-targeting NAM, it should be underlined that this proof-of-concept compound presents a starting platform given its apparently low potency. Future efforts are required to deepen our understanding of how to improve affinity and potency, which will be essential for considering therapeutic applications. Structural differences between compounds with and without effect, as well as the data from the MD simulations with C407 and non-binder C476, suggest that a polar moiety, such as an amide, which can form polar interactions with $S351^{3x40}$, might be beneficial for an effective compound. However, it will require more tested compounds and further investigations to understand how to improve affinity and potency. Furthermore, it would be of interest to achieve FZD paralog selectivity, although this might be hindered by the overall sequence conservation of FZDs in the region of the potential binding site. Here, it generally seems more likely to achieve selectivity for a subgroup of FZDs than a specific FZD paralog (see also Supplementary Table 1)[7]. Nevertheless, mutagenesis confirmed the proposed binding site, and cryo-EM data revealed a density within the 7TMD of $FZD_7$. Although this density is consistent with the predicted location of C407, it was not possible to unambiguously assign it to the compound. The uncertainty in this density can in this case be attributed to either low occupancy due to the low affinity of C407 to bind to $FZD_7$ or to multiple binding poses of the compound within the binding site (as also suggested by the MD simulations). While the assignment of C407 to this density should be interpreted with caution, the prospective pose of C407 is supported by functional data and MD simulations. Hence, these data provide potentially cryo-EM evidence for small molecule interaction within the 7TMD of a FZD.

MD simulations starting from the docking pose, as well as the pose modeled based on the cryo-EM data, confirm this further. In both cases, C407 behaves similarly with a movement deeper into the 7TMD core of the receptor, where it reaches the tight base of the internal water cavity[16]. The differences in the observed final poses can likely be attributed to the different starting conformations of the receptor in each case, i.e., an active conformation compared to an inactive conformation, which impacts side chain orientations and flexibility of the receptor. Similarly, the divergent location of the density potentially corresponding to C407 in the cryo-EM data can be attributed to a receptor conformation that is in an inactive state and especially stabilized as a non-natural homodimer. We assume that the movement of the compound within the receptor is hindered by the rigidity of the inactive, antiparallel dimer, which corresponds to the observations from the MD simulations and the cryo-EM data. Importantly, both the initial poses as well as the deeper pose after MD simulations could be confirmed by site-directed mutagenesis of residues forming polar interactions with C407 in the respective poses ($Y489^{6x51}$ and $S351^{3x40}$, respectively). Thus, we carefully suggest a potential binding mechanism, where the compound enters the 7TMD of $FZD_7$ from the extracellular side and finally moves further down to a stable binding pose, from which it exerts its effect. This suggested binding mechanism is also in line with previous observations that suggest a dynamic opening and closing of this cavity to the extracellular side[16]. However, further investigations to confirm this would be required, considering, for example, the flexibility of the CRD impacting the ligand entry due to occlusion effects. Furthermore, the MD simulations revealed a potential mechanism of action of the NAM C407, involving the conformational rearrangement of residues $W354^{3x43}$ and $Y478^{6x40}$, which are part of an extended molecular switch involved in the activation of

FZDs[6,18]. In this context, the question arises why the compound density detectable by cryo-EM in the inactive FZD$_7$ is not positioned at the bottom of the internal cavity in the mechanistically most relevant pose. We can only speculate that the inactive antiparallel dimer is too rigid, which would not allow subtle side chain rearrangements to accommodate C407 at the same deep pose that was revealed by the MD simulations. Surely, the characterization of binding poses and the description of a potential mechanism of action will serve to further develop additional FZD-targeting small molecule modulators of WNT-signaling. Conceptually, it is remarkable that the binding site and potential mechanism of action of a FZD-targeting NAM can be linked to receptor activation networks through the highly conserved residues Y$^{6x51}$ and extended molecular switch Y$^{6x40}$ and W$^{3x43}$. This is in line with the recent identification of a conserved molecular switch, the extended molecular switch and state-stabilizing residues apparently important for FZD activation[6,7,17,18].

In summary, C407 serves as a proof-of-concept showing that the design of small molecule modulators targeting the 7TMD of FZDs is, in principle, possible, mechanistically feasible as well as, efficient to target WNT/$\beta$-catenin signaling that is seen as a driver in pathologies such as cancer[30]. While C407 interacts with conserved residues in the FZD 7TMD core and acts as a NAM, it is by no means an optimal ligand and further improvements are necessary for it to be of use—it does, however, represent a suitable starting point for future developments. Targeting the FZD core binding site and the suggested mechanism of action in specific, might result in improved FZD-targeting ligands and modulators in the future.

## Methods
### In silico methods
**Preparation of docking setups.** All docking setups were based on the FZD$_7$ structure with PDB ID 7EVW (now 8YY8)[6]. The G protein, as well as any additional molecules, were removed prior to further structure preparation. Missing side chains in this structure were added using the Dunbrack rotamer library in UCSF Chimera (v.1.16)[31,32] and the receptor was protonated. Residue H303$^{2x58}$, which is located within the 7TMD and in proximity to a potential ligand binding site, was protonated in $\epsilon$-position to allow for polar interactions with a potential ligand. To define potential small-molecule binding sites within the 7TMD of the receptor, the FZD$_7$ structure was aligned to SMO structures and the respective ligand was copied to the FZD$_7$ structure. To resolve clashes, the obtained complexes were then minimized using the CHARMm22 force field[33] and including the template ligand and all residues in 5 Å distance to the molecule as well as any added side chains and all hydrogen atoms in this minimization. Of note, residue R496$^{6x58}$ ended up slightly aplanar in several of these minimized structures, however, within experimentally observed angle distributions for this side chain[34] and was hence accepted. The resulting receptor and template ligand conformations were then prepared for DOCK3.7, ensuring that the grids and spheres used by DOCK to place the molecules in the binding site were constrained to the center of the 7TMD.

Finally, four docking setups were obtained: $^{sag}$FZD$_7$ (template ligand based on SAG1.5 from SMO structure 6XBL[35]; the ligand was slightly modified by removing the pyridine); $^{cyclo}$FZD$_7$ (template ligand based on cyclopamine from SMO structure 4O9R[36]); $^{vis}$FZD$_7$ (template ligand based on vismodegib from SMO structure 5L7I[37]; ligand was slightly modified containing a benzyl-moiety instead of a pyridyl-moiety); and $^{sant}$FZD$_7$ (template ligand based on SANT-1 from SMO structure 4N4W[38]).

**Docking calculations.** The ZINC15[39] drug-like (9,145,528 molecules and protomers) and lead-like (2,783,169 molecules and protomers) libraries were docked to all four docking setups using DOCK3.7[40]. The top 500 ranked molecules were then inspected visually to account for artifacts and artificially inflated scores from the scoring function.

Finally, a total of 22 molecules were selected to be purchased and tested in cell-based assays.

**Secondary screens.** After identifying C45 as an initial hit, secondary screening campaigns were used to identify more similar hit molecules with potentially improved properties. For this, first, a new docking setup based on C45 was prepared. Here, the docking pose of C45 was used as a template ligand in conjunction with the receptor conformation from its original docking ($^{sant}$FZD$_7$) to create the grids and spheres for DOCK. This docking setup will be referred to as $^{c45}$FZD$_7$.

In an initial follow-up screen, a molecular library by the Chemical Biology Consortium Sweden (CBCS) containing 135,545 molecules was used. The library was filtered for all molecules with an ECFP4 Tanimoto similarity above 0.4 to C45, resulting in a subset of 342 molecules. This subset of the CBCS library was then docked to setup $^{c45}$FZD$_7$ using DOCK3.7. Resulting docking poses were evaluated visually, and 11 molecules were selected to be tested in cell-based assays.

After identifying the additional hit C407, a second follow-up screen based on this molecule was conducted. Molecules similar to C407 were retrieved from the Enamine screening compound catalog, resulting in a focused library of 177 molecules. This library was then docked to the docking setup $^{c45}$FZD$_7$ using DOCK3.7, and after visual evaluation of the resulting docking poses, eight molecules were selected to be purchased and characterized in cell-based assays.

**MD simulations. Structure preparation and MD system setup.**
Four MD systems were set up for the MD simulations: one starting from the docking pose of C407 and using the receptor structure as prepared for the corresponding docking setup ($^{sant}$FZD$_7$/$^{c45}$FZD$_7$ called "docking pose"); one using the same receptor structure but without a ligand bound ("*apo* active conformation"); one starting from the docking pose of C476 with the same receptor structure as described above ("non-binder C476"); and one based on the receptor structure modeled in the cryo-EM density and including C407 ("cryo-EM pose"). Prior to full MD system setup, these structures were prepared for MD simulations to close structural gaps and similar.

The receptor structure used for the MD simulations starting from the C407 and the C476 docking pose, as well as for the *apo* active conformation MD simulations, was prepared as described in the following. Missing parts of extracellular loop 3 (ECL3) were added using the AlphaFold model as deposited in the AlphaFold database[41,42], copying the orientation of residues R501 to F527. Furthermore, some parts of the original receptor structure were in disagreement with the linked cryo-EM data and were hence replaced with the respective regions from a remodeled structure (PDB ID 9EW2)[16]. These regions were C210 to P233 (linker domain), V316 to G335 (ECL1), R451 to S475 (intracellular TM5 to TM6) and T543 to T553 (intracellular TM7 to parts of H8). Furthermore, the rotamers of residues R238, H498, W499, W503, and V513 were adapted to resolve clashes introduced by these changes.

For the MD simulations starting from the cryo-EM pose, only the dimer subunit containing the density and the modeled pose of C407 was used for MD simulations. Missing loops that could not be modeled based on the cryo-EM density were added using Coot (0.9.8.95).

Force field parameters for C407 and C476 were described with GAFF2[43], an improved version of the General AMBER Force Field[44]. Following the procedure described in ref. 43, the topologies were generated by first optimizing the geometry of the molecules at the HF/6-31G$^*$ level of theory using Q-Chem[45]. The geometries were then passed to ACPYPE[46], a Python script that uses Antechamber[44,47] and facilitates the generation of GAFF2 molecular topologies.

The MD simulation systems and input were then generated using the CHARMM-GUI bilayer builder[48,49]. The receptor models were oriented by pre-aligning them to the entry for PDB ID 7EVW in the Orientations of Proteins in Membranes database using UCSF Chimera

(v.1.16)[32,50]. Histidine protonations were assigned as used for the docking calculations and all disulfide bonds were assigned as pre-defined (C210-C230; C234-C315; C336-C411; C508-C515). Termini of the receptor were patched by acetylation and methylamidation. The receptor was placed within a hexagonal box in a bilayer of palmitoyl-oleoyl-phosphatidylcholine (POPC). To neutralize the system, Na$^+$ and Cl$^-$ ions were added up to a concentration of 150 mM. Finally, the system was solvated using TIP3P water. Ligand force field parameters were replaced with the ones derived as described above. A summary of system setup parameters can be found in Supplementary Table 3.

**MD simulations**. The aim of the MD simulations was to assess the stability of the binding pose and interactions of C407 within its binding site in the 7TMD core of FZD$_7$. For this purpose, all-atom simulations with explicit solvent are suitable to describe the desired observables.

All MD simulations were run using GROMACS 2024 and AMBER force fields (protein: FF19SB; lipids: Lipid21; water: TIP3P), which are suitable to describe membrane protein systems and interactions with small molecules[51–53]. Initially, the MD system was energy minimized with positional restraints using steepest descent until the maximum force was below 1000 kJ mol$^{-1}$ nm$^{-1}$. This was followed by 250 ps equilibration in an NVT ensemble and 1750 ps equilibration in an NPT ensemble while stepwise loosening the positional restraints on atoms in the system. Initial velocities were randomly assigned based on the Maxwell distribution at 310 K. A last equilibration step of 15 ns and with only light positional restraints (50 kJ mol) on protein backbone and ligand heavy atoms was run in three replicates and with new random velocities assigned for two of these to ensure that observed results were independent from starting conditions. Finally, production runs of 300 ns (time step: 2 fs) were performed in three replicates starting with the velocities from the last equilibration step and without any posi-tional restraints. This timescale is sufficient to investigate the stability of a bound compound within the binding site.

During both equilibration and production the temperature of 310 K and the pressure of 1 bar were maintained using the v-rescale and the c-rescale algorithms, respectively. Bonds to hydrogen atoms were constrained using the LINCS algorithm[54]. Long-range non-bonded interactions were cut off at 0.9 nm and electrostatic interactions were calculated using the Particle-Mesh Ewald algorithm[55].

MD parameter files for minimization, equilibration and produc-tion as well as initial and final configurations of the systems can be found in Supplementary Data 5. The MD simulation checklist can be found in Supplementary Data 6.

**MD analysis**. Production simulations were centered and aligned using GROMACS 2024[51]. The entire 300 ns of each production run were included for analysis. Measurements such as RMSD, RMSF, dis-tances and dihedral angles, as well as clustering of frames, were per-formed using AmberTools 18 CPPTRAJ[56]. Frame clustering based on C407 poses can be found in Supplementary Data 4. Interactions between ligand and protein were computed and plotted using get-contacts (https://getcontacts.github.io/). All MD trajectories are deposited on GPCRmd (https://gpcrmd.org/dynadb/publications/1629/) and individual simulation IDs 2339 (docking pose), 2340 (*apo* active), 2341 (structure pose) and 2364 (C476)[57].

## In vitro methods

**Ligand preparation**. WNT-3A (Biotechne, cat.-No.: 5036-WN) was obtained as a lyophilized product and resuspended in Dulbecco's Phosphate-Buffered Saline (DPBS) containing 0.1% bovine serum albumin (BSA) at a concentration of 100 µg/mL. A vial of resuspended WNT-3A was kept for a maximum of 8 weeks between 4 and 8 °C. Vehicle controls were prepared for each batch according to the data sheet obtained from the manufacturer.

**Cell culture and transfection**. HEK293A cells (Thermo Fisher Scien-tific, cat.-No.: R70507) and *ΔFZD$_{1-10}$* cells (kind gift from Benoit

Vanhollebeke, Université de Bruxelles) were routinely maintained in Dulbecco's Modified Eagle's Medium (DMEM), which was supple-mented beforehand with 10% fetal calf serum (FCS) and 1% penicillin/streptomycin (Gibco, cat.-No.: 151-40122) in a humidified incubator (5% CO$_2$). Cell culture plastics to maintain the cell culture were purchased from Sarstedt unless stated otherwise.

Cells were generally transfected with 1 µg of DNA per mL cell suspension using linear polyethyleneimine (PEI Max, Polysciences Inc., stock concentration: 1 mg/mL) as the transfection reagent at a PEI:DNA ratio of 3:1 (µL:µg). Plasmid percentages indicated in the respective subsections below refer to the percentage of the total transfected plasmid amount. Empty pcDNA3.1 was added when necessary to bal-ance the transfected DNA amount to 1 µg per mL cell suspension. Cells were regularly tested for mycoplasma contamination and tested negative.

**BRET-based competition binding assays**. HEK293A cells (350,000 cells/mL) were transfected in suspension with 0.5% Nluc-FZD$_7$ or HiBiT-FZD$_7$ and 35,000 cells were seeded into poly-D-lysine-coated white, opaque 96-well plates (Greiner Bio-One). Two days after transfection, the medium was removed, the cells were washed once with Hank's balanced salt solution (HBSS), and 80 µL of HBSS was added to the wells. Next, 10 µL of 10× compound dilution or DMSO (in HBSS) was added, followed by the addition of 10 µL of BODIPY-cyclopamine (final concentration: 200 nM). After 90 min incubation time at 37 °C (no additional CO$_2$), furimazine (for Nluc-FZD$_7$, diluted 1/1000 in HBSS; Nano-Glo substrate, Promega, cat.-No.: N1110) or a mix of LgBiT and furimazine (for HiBiT-FZD$_7$, both diluted 1/200 in HBSS, Promega, cat.-No.: N2421) was added, the plate was incubated for 15 min in the dark and BRET was measured five times using a TECAN Spark microplate reader with the following settings: Nluc bioluminescence was filtered between 445 and 470 nm, fluorescence originating from the fluor-escent ligand (BODIPY-cyclopamine) was detected between 520 and 545 nm.

**TOPFlash reporter gene assay**. To assess the effect of selected compounds on WNT/β-catenin-dependent signaling, *ΔFZD$_{1-10}$* HEK293T cells (500,000 cells/mL) were transiently transfected (with a mix of 30% of HiBiT-FZD$_7$, 25% 8X SuperTOPFlash (Fluc, Addgene No. 12456) and 5% of Renilla control plasmid (pRL-TK, Promega)) and seeded (100 µL per well) into a PDL-coated, white opaque 96-well plate. After 20–24 h, the medium was removed and replaced with starvation DMEM (no FCS) containing the indicated ligands/DMSO and WNT-3A/vehicle. After an additional incubation period of 20-24 h inside the incubator (5% CO$_2$), the cells were washed once with HBSS and the measurement was conducted on a TECAN Spark microplate reader using the Dual-Luciferase Assay System (Promega, cat.-No.: E1910) following a slightly modified protocol[17,26]. The pathway-activity-dependent Fluc bioluminescence was detected between 550 and 620 nm, while Rluc bioluminescence (control for transfection effi-ciency) was detected between 445 and 530 nm.

**Control assays for TOPFlash**. Different experimental setups were designed to validate the results from TOPFlash experiments. Non-specific interference with Fluc bioluminescence was assessed by con-stitutively expressing solely the Fluc plasmid at 0.5% (adjusted to 1 µg per mL suspension with pcDNA3.1) in a PDL-coated, white opaque 96-well plate. After 24 h, the medium was removed and replaced with serum-free DMEM containing 10 nM C59 and 50 µM of compounds. After another 24 h, the medium was removed, washed once with HBSS and 20 µL of 1× passive lysis buffer was added and incubated for 15 min at RT with gentle shaking. Subsequently, 20 µL of LARII reagent was added and Fluc bioluminescence was read independent of any FZD. To prove that the C407-induced reduction of TOPFlash responses is due to actual binding to FZDs, DVL2 overexpression was used to induce

TOPFlash responses in the absence of any FZD. To do so, $\Delta FZD_{1-10}$ HEK293T cells (500,000 cells/mL) were transiently transfected (with a mix of 10% of FLAG-hDVL2 (kindly provided by Vitězslav Bryja), 25% 8X SuperTOPFlash and 5% of pRL-TK) and seeded on PDL-coated white 96-well plates. After one day of incubation, cells were treated with compounds/DMSO dissolved in starvation DMEM and incubated for an additional day. The measurement was conducted on a TECAN Spark microplate reader using the dual-luciferase assay system as described above.

**DEP-Clamp conformational BRET assays.** To monitor the effect of C407 on the conformation rearrangement of FZD and the DEP domain of DVL2 induced by WNT-3A stimulation, we employed a unimolecular BRET biosensor setup, referred to as FZD-DEP clamps[21]. These sensors consist of FZD, Nluc, a 10 nm E/RK linker, mVenus, and the DEP domain of DVL2, which constitutively binds tightly to FZDs. When treated with WNT-3A, a conformational change of the FZD-DEP-complex leads to a rearrangement of Nluc and mVenus, resulting in a positive BRET change. HEK293A cells were transfected in suspension with 2% FZD-DEP-Clamp plasmid and 98% pcDNA 3.1 per mL of cells at a density of 350,000 cells/mL using a 3:1 w/w excess of PEImax (Polysciences). Mutations to the wild-type $FZD_7$-DEP clamp sequence were introduced with the GeneArtTM Site-Directed Mutagenesis System (Thermo Fisher #A13282) according to the manufacturer's instructions. Transfected cells were transferred to an opaque white 96-well plate (Greiner BioOne) at a density of 35,000 cells per well. After an incubation period of 40–48 h, media was removed, cells were washed once in HBSS, and 70 µL of HBSS + 0.1% bovine serum albumin (BSA) were added to each well. Then, cells were treated with 10 µL of a 1 mM C407 stock (dissolved in DMSO and diluted in HBSS + 0.1% BSA) to reach a final concentration of 100 µM or 50 µM, or with a DMSO equivalent (1% final DMSO concentration) for 20 min at 37 °C. Afterwards, 10 µL of furimazine (final concentration 1:1000 in HBSS + 0.1% BSA; Promega #N1120) were added to the cells and, after an incubation period of 6 min, BRET was read three consecutive times on a TECAN Spark multimode plate reader (Nluc emission: 445–485 nm, mVenus emission: 520–560 nm; integration time: 100 ms). Then, WNT-3A (or vehicle control) was added to a final concentration of 1000 ng/mL, and BRET was measured for 60 min in 2 min intervals. The raw BRET ratio was obtained by dividing Nluc counts by mVenus counts. The raw BRET ratio was then first baseline-corrected for the initial three reads and secondly, WNT-3A-treated samples were corrected for vehicle-controlled samples to calculate $\Delta BRET$ values. Average $\Delta BRET$ values were obtained by taking the mean of all data points from 2 to 60 min after stimulation. For each biological replicate, average $\Delta BRET$ values of C407-treated samples were normalized to DMSO-treated samples to calculate the effect of C407 on WNT-3A-induced conformational changes at the FZD-DEP interface. Significance was assessed by one-way analysis of variance followed by Dunnett's post-hoc test or multiple unpaired, two-tailed t-tests, as detailed in the respective figure legends.

**Cell viability.** HEK293A cells were seeded at a density of 1000 cells per well in complete DMEM media in a black opaque 96-well plate (Greiner BioOne). One day later, the medium was exchanged with fresh media supplemented with the tested compounds at a final concentration of 50 µM or DMSO (vehicle control). Three days later, the medium was exchanged with fresh DMEM supplemented with the tested compounds following the previous protocol. After three more days, or one week after initial seeding, the medium was removed and 90 µL of fresh DMEM was added with 10 µL of AlamarBlue HS reagent (Thermo Fisher Scientific). The cells were incubated for 4 h at 37 °C inside an incubator (37 °C, 5% $CO_2$) then fluorescence was read using a TECAN Spark multimode microplate reader (excitation: 535 ± 25 nm, emission: 595 ± 35 nm). For cell viability assays, the fluorescence in the ligand-treated wells was normalized to wells treated with DMSO.

## Biochemistry and structural methods

**Expression and purification of $FZD_7$.** Full-length $FZD_7$ was purified as before. Briefly, $FZD_7$ was expressed in Sf9 insect cells (Expression System) using the Bac-to-Bac baculovirus expression system (ThermoFisher). Cell cultures were grown in EX-CELL 420 Serum-free medium to a density of $2 \times 10^6$ cells/mL then infected with baculovirus at a volume ratio of 1:50. For the last hour of purification, C407 was added at a concentration of 1 µM. Then cells were harvested after 48 h by centrifugation at $4000 \times g$ for 15 min then kept frozen at −20 °C for further use.

In order to enhance binding of C407 to $FZD_7$, C407 was added to all buffers used during purification at a concentration of 1 µM and 10 µM for detergent extraction. Cell pellets were thawed and resuspended in lysis buffer containing 10 mM TRIS-HCl pH 7.5, 100 mM NaCl, 1 mM EDTA, and protease inhibitors [leupeptin (5 µg/mL) (Sigma Aldrich), benzamidine (10 µg/mL) (Sigma) and phenylmethylsulfonyl (PMSF) (10 µg/mL) (Sigma Aldrich)] then centrifuged at $4000 \times g$ for 15 min. Supernatant was removed and then pellets were resuspended and homogenized using a glass Dounce tissue grinder (10 strokes with A pestle and 20 strokes with B pestle) in solubilization buffer containing 50 mM TRIS-HCl pH 8, 200 mM NaCl, 1% LMNG (Anatrace), 0.1% CHS (Sigma Aldrich), 0.1% GDN (Anatrace) iodacetamide (2 mg/mL) (Anatrace), protease inhibitors, and 1 µM C407. The mixture was stirred for 2 h at 4 °C, then centrifuged for 30 min at $38,400 \times g$. The cleared supernatant was incubated with 2 mL Strep-Tactin resin (IBA) for 2 h at 4 °C. The resin was washed with 10 column volumes (CVs) of high salt buffer containing 50 mM TRIS-HCl pH 7.5, 500 mM NaCl, 0.02% LMNG, 0.002% CHS, 0.002% GDN, and 1 µM C407 followed by 15 CVs of low salt buffer (same as high salt buffer but with 100 mM NaCl). $FZD_7$ was eluted with the same buffer containing 2.5 mM desthiobiotin (IBA) and samples corresponding to the dimeric protein were concentrated in a 50-kDa molecular weight cutoff concentrator (Millipore) to 7.8 mg/mL prior to loading onto a size exclusion column.

A Superdex 200 Increase 10/300 GL, GE Healthcare column was equilibrated with buffer containing 10 mM TRIS-HCl pH 7.5, 100 mM NaCl, 0.002% LMNG, 0.0002% CHS, 0.0002% GDN, and 1 µM C407 and $FZD_7$ was injected. Peak fractions corresponding to $FZD_7$ were pooled and concentrated to 2.88 mg/mL.

**Cryo-EM methods.** The purified $FZD_7$-dimer sample was supplemented with C407 at a molar ratio of 1:10. Next, a 3 µL sample was applied to a glow-discharged (20 mA, 40 s.) UltrAuFoil R 1.2/1.3 300-mesh copper holey carbon grid (QuantiFoil, Micro Tools, GmbH, Germany), blotted for 2.5 s, then flash-frozen in liquid ethane using a Vitrobot Mark IV set at 4 °C and 100% humidity (Thermo Fisher Scientific). Images were collected on a Titan Krios G3i operated at 300 kV at the SciLifeLab Solna Campus. Micrographs were recorded using a Gatan K3 BioQuantum detector in super-resolution mode using EPU software (v2.14.0). A total of 19,899 movies were obtained at a magnification of 165,000 corresponding to a 0.507 Å calibrated pixel size and exposure dose of 80.1 e/Å with defocus ranging from −0.4 µM to 1.8 µM.

**Data processing.** Data processing for the $FZD_7$-dimer with C407 dataset was performed using cryoSPARC (v4.7.1). Movie frames were aligned using Patch Motion Correction and Contrast Transfer Function (CTF) parameters were estimated by Patch CTF correction. Particle picking was done by automatic Gaussian blob detection (mask diameter = 140 with elliptical and circular blob) yielding particles that were subjected to reference-free 2D classifications (classes = 100, mask diameter = 150). Particles were Fourier cropped to a box size of 360 px and downsampled to 90 px corresponding to 2 Å/px. Iterative 2D classifications identified 806,426 particles that were used as references to train Topaz (v0.2.5a), a positively unlabeled convolutional neural network for particle picking. Topaz picked approximately 6.7 million

particles that were input into further 2D classifications. Particles were selected from the best 2D classes (474,426 particles) and launched into multiple rounds of Ab-initio reconstruction (C1 symmetry) with 6 classes. The classes comprising the best dimeric reconstructions were further launched into 2D classifications and a subset of 180,361 were selected and re-extracted with a pixel size of 1.01 Å and used for non-uniform refinement, local and global CTF refinements, higher order aberrations correction, and symmetry expansion. After iterative refinements, the final subset of particles yielded a map with an overall resolution of 2.48 Å. To deconvolute the conformational hetero-geneity of the dimer, the particles were exported using csparc2star.py to Relion-5.0 to generate a loose mask to use as a focused mask on the dimer. The mask was used for 3D classification without alignment (class similarity = 0.01) with the symmetry expanded particles, which were then used for local refinement. Focused classification resulted in two classes, the first at 3.28 Å and the second at 2.50 Å. The second class was used as the final map for model building. The previously solved FZD$_7$-dimer was used as the starting model and fit into the experimental cryo-EM density map using UCSF Chimera (v1.17.3). Comparison of the previously solved structure with the model revealed a slight compression of the map, which has been observed before due to differences in microscopes that vary in the defocus-dependent beam tilt. Although beamtilt and per-particle-magnification correction were performed, this ultimately did not ameliorate the issue. Consequently, the pixel size was recalibrated by improving the cross-correlation coefficient (CC) of the refined model to an overall CC of 0.65, resulting in a pixel size of 1.045. The model in the adjusted reconstruction was iteratively refined using global minimization in Phenix (1.21) and real-space-refinement in Coot (0.9.8.95). Restraints for cholesterol hemisuccinate and C407 were generated using the Ligand Validation Server (https://grade.globalphasing.org/cgi-bin/grade2_server.cgi), docked using Coot, and refined in Phenix. Final-map-model validations were carried out using MolProbity in Phenix. The overall statistics are reported in Supplementary Table 2.

### Preparation of figures
Figures were designed using Chimera (v.1.16)[32], ChimeraX (v.1.10)[58], gnuplot, matplotlib and GraphPad Prism 10 (GraphPad Prism Software Inc.). Assay schemes were created with BioRender.com.

### Materials
**Test compounds.** All compounds selected from the primary screen were purchased from either MolPort or Enamine, with a single compound ordered from Santa Cruz Biotechnology. The second series of compounds was taken from the CBCS library. The third round of compounds was ordered from Enamine. All compound identifiers and vendor information, as well as purities as given by vendor quality control, can be found in Supplementary Table 5, and SMILES for all tested compounds are listed in Supplementary Table 6.

### Reporting summary
Further information on research design is available in the Nature Portfolio Reporting Summary linked to this article.

## Data availability
All relevant data generated and analyzed during this study are included in this article and its Supplementary Information. Source data and reporting summary are provided with this paper. CryoEM data are available as the PDB entry pdb_00009rhg (PDB ID 9RHG; https://doi.org/10.2210/pdb9RHG/pdb) and EMDB entry ID EMD-53969. All MD trajectories are deposited on GPCRmd (https://gpcrmd.org) with publication entry https://gpcrmd.org/dynadb/publications/1629/and individual simulation IDs 2339 (docking pose), 2340 (apo active), 2341 (structure pose) and 2364 (C476). Should any raw data files be needed in another format, they are available from the corresponding author upon request. Expression vectors used and created for this work can be obtained from the corresponding author. Source data are provided with this paper.

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

## Acknowledgements

We thank Benoit Vanhollebeke for the $\Delta FZD_{1-10}$ HEK293 cells. The work was supported by grants from Karolinska Institutet, Cancer Research KI Translational Seed Funding Grant Agreement (project code 221), the Swedish Research Council (GS: 2017-04676, 2019-01190, 2024-02515), the Swedish Cancer Society (GS: CAN2017/561, 20 1102 PjF, 23 2825 Pj), the Swedish Society for Medical Research (MMS: PG-22-0379), the Novo Nordisk Foundation (GS: NNF21OC0070008, NNF22OC0078104), The German Research Foundation (DFG; LG: 504098926; MMS: 470002134; JHV: 520506488), and the SciLifeLab & Wallenberg Data Driven Life Science Program (grant: KAW 2020.0239). DCB acknowledges funding from the European Union's Horizon Europe research and innovation programme under the Marie Skłodowska-Curie grant agreement No. 101199012. The MD simulations were enabled by resources provided by the National Academic Infrastructure for Supercomputing in Sweden (NAISS; NAISS 2023/5-419, NAISS 2024/5-524, NAISS 2024/22-695), partially funded by the Swedish Research Council through grant agreement no. 2022-06725.

## Author contributions

Conceptualization: M.M.S. and G.S.; methodology: all authors; data curation, formal analysis, and visualization: M.M.S., L.G., J.K., J.H.V., D.C.-B., and B.F.; funding acquisition, project administration, and resources: G.S. and P.K.; investigation: M.M.S., L.G., J.K., and J.H.V.; writing-original draft: M.M.S., L.G., J.K., and G.S.; writing-review & editing: M.M.S., J.K., and G.S. with contributions from all authors.

## Funding

## Competing interests

The authors declare no competing interests.
