## [Transparent Peer Review file · Nature Communications]

In silico docking yields small molecule negative allosteric modulators targeting the core of Frizzled 7

Corresponding Author: Professor Gunnar Schulte

A version of this paper was originally rejected for publication by Nature Communications, however that decision was reconsidered after appeal by the authors.

Version 1:

Reviewer comments:

Reviewer #1

(Remarks to the Author)

Magdalena M. S. and co-workers described the identification and characterization of C407, a small-molecule negative allosteric modulator targeting the 7-transmembrane domain (7TMD) of Frizzled 7 (FZD₇). A multi-disciplinary approach, including in silico docking, pharmacological assays, site-directed mutagenesis, cryo-electron microscopy (cryo-EM), and molecular dynamics (MD) simulations, was used to validate C407's binding site and mechanism of action in inhibiting WNT/β-catenin signaling. The study offers a valuable proof-of-concept for targeting FZD7 7TMD, distinct from canonical WNT/CRD interactions. However, some points warrant further clarification or expansion.

1. C407 exhibits modest potency (pIC₅₀=4.86), and efforts to optimize it yielded no improvements. A more detailed analysis of SAR, including why specific modifications failed to enhance potency, could guide future hit optimization. To strengthen this, docking/MD analysis of inactive compounds should be included, and key pharmacophoric features should be confirmed via energy decomposition (e.g., MM/GBSA).

2. For rigor, it would be preferable to first perform MD refinement of the C45–FZD7 complex, then use representative structures for subsequent focused screening. This may improve the identification of higher-affinity analogs.

3. While in vitro and biosensor data support C407's activity, its efficacy in disease models (e.g., intestinal tumor organoids or xenografts) is not addressed.

4. C407 shows similar activity across FZD4/5/7. The lack of paralog selectivity should be discussed in terms of therapeutic implications, and structural/sequence comparisons or mutagenesis studies could be conducted to explore the feasibility of achieving selectivity.

5. While TOPFlash data suggest FZD7 as the primary target, the possibility of C407 interacting with other WNT/β-catenin pathway proteins is not excluded.

6. The proposed ligand entry route via the extracellular side into the 7TMD core warrants more detailed MD-based pathway sampling or adaptive biasing simulations.

7. Since apo-FZD7 was proposed to stabilize microswitches (e.g., W354^{3×43}, Y478^{6×40}), it is recommended to include comparative MD simulations starting from the 9EPO apo structure to directly quantify differences in residue rigidity and interaction networks. In addition, the author claimed that ligand-bound cryo-EM conformations are more rigid than apo models could be substantiated by providing RMSF values for key residues.

8. The final C407 binding poses originating from the docking and cryo-EM conformations diverge by 4–6 Å. The manuscript should better explain the observed instability or shift of the cryo-EM pose and how receptor conformation modulates ligand binding.

Reviewer #2

(Remarks to the Author)

In this study, Dr. Scharf employed molecular docking to identify small molecules targeting the transmembrane core domain of FZD7, followed by validation using functional assays, site-directed mutagenesis, molecular dynamics simulations, and cryo-EM analysis. The seeking of small-molecule binders for FZDs is novel, given the current lack of validated ligands for this receptor family, which has significantly impeded the development of FZD-targeted anti-cancer therapies. However, several concerns need to be addressed before the manuscript can be considered for publication.

1. The density corresponding to the C407 ligand appears weak. The authors should apply cryo-EM single particle analysis techniques, such as focused classification or local refinement etc., to better resolve and confirm the presence of C407 within the receptor structure.
2. It would be valuable to assess the specificity of C407 across other FZD subtypes, such as FZD6, FZD2, and FZD4.
3. The authors propose that C407 functions as a negative allosteric modulator (NAM) of FZD7 by inhibiting DVL recruitment. It would be informative to compare the structure of C407-bound FZD7 with the recently published FZD4–DEP complex, which may provide mechanistic insights into how C407 modulates receptor activity.

Reviewer #3

(Remarks to the Author)

The study by Scharf et al. reports the discovery of C407, the first-in-class negative allosteric modulator of FZD7 that acts by binding to its 7TM domain. This is a rigorous computational, structural, and molecular pharmacology study employing compound docking, in silico library screening, pharmacological assays with BRET- and luminescence-complementation-based conformational and signaling biosensors, site-directed mutagenesis, cryo-EM structure determination, and MD simulations. The authors demonstrate great attention to details, possible nonspecific mechanisms of action (e.g. luciferase inhibition), and alternative interpretations of low-resolution data (e.g. cryo-EM densities).

Unfortunately the presented molecule, C407, is weak and non-selective, and limited efforts of optimizing it via SAR-by-catalog did not lead to more potent NAMs. Nevertheless, this molecule can serve as a proof-of-principle for the proposed model of modulation of receptors in the FZD family, complementing the previously characterized inhibitors of SMO that act through the same mechanism.

The authors did an excellent job addressing potential caveats of the study and employing a combination of experimental and computational approaches to elucidate the mechanism of action of C407. I only have two minor comments.

* The authors made compound-bound structures and models available for review, which is commendable. Encouragingly, they also deposited the MD trajectories to <https://gpccrmd.org/>; however, no access code was provided to the reviewers. I encourage the authors to include this info with the final published paper.

* I recommend adding a schematic of the BODIPY-Cyc competition binding assay to Fig 1 and a schematic of the TOPFlash to Fig 2, similar to that of FZD-DEP/Clamp biosensor in Fig 2C.

Version 2:

Reviewer comments:

Reviewer #1

(Remarks to the Author)

After a comprehensive review of the manuscript, I have no further questions or concerns.

Reviewer #2

(Remarks to the Author)

I am ok with the answers to my questions.

Reviewer #3

(Remarks to the Author)

This is a revised version of the study by Scharf et al. that reports the discovery of C407, the first-in-class negative allosteric modulator of FZD7 that acts by binding to its 7TM domain. The authors did an excellent job addressing reviewers' comments and critiques for the initial submission, including the addition of new analyses and MD simulations. My own comments are addressed in full. I believe the article, and the compound, would become great resources for the pharmacology and drug discovery community.

REBUTTAL LETTER – author’s comments in green (green text in **bold** emphasizes changes that have been added to the manuscript during the revision)

Reviewer’s Comments:

Reviewer #1 (Remarks to the Author)

Magdalena M. S. and co-workers described the identification and characterization of C407, a small-molecule negative allosteric modulator targeting the 7-transmembrane domain (7TMD) of Frizzled 7 (FZD₇). A multi-disciplinary approach, including in silico docking, pharmacological assays, site-directed mutagenesis, cryo-electron microscopy (cryo-EM), and molecular dynamics (MD) simulations, was used to validate C407’s binding site and mechanism of action in inhibiting WNT/β-catenin signaling. The study offers a valuable proof-of-concept for targeting FZD7 7TMD, distinct from canonical WNT/CRD interactions. However, some points warrant further clarification or expansion.

We thank the reviewer for the appropriate summary and the appreciation of the proof-of-principle that our work provides.

1. C407 exhibits modest potency ($pIC_{50}=4.86$), and efforts to optimize it yielded no improvements. A more detailed analysis of SAR, including why specific modifications failed to enhance potency, could guide future hit optimization. To strengthen this, docking/MD analysis of inactive compounds should be included, and key pharmacophoric features should be confirmed via energy decomposition (e.g., MM/GBSA).

We agree that a detailed SAR analysis could be informative to guide future hit optimization. However, since we only investigated a relatively small number of compounds (19 compounds with a similar core scaffold, of which 10 were characterized with higher confidence) and the ligands among these showed only small differences in IC_{50} values, it is not possible to perform such a detailed analysis. Hence, we did not perform additional analyses such as energy decomposition calculations of pharmacophoric features. Therefore, we have only included a cautious analysis into the manuscript and would prefer to keep it this way. The respective paragraph that was already included in the results section is the following:” *All of the additionally tested molecules as well as the initial hit molecules C45 and C407 contain a similar structural core motif (5-methyl-3H-thieno[2,3-d]pyrimidin-4-one). While it is difficult to derive a structure-activity-relationship due to the small set of compounds and the small differences in pIC_{50} values, a few trends can be observed. Specifically, compounds that contain an ethyl-ester moiety as well as an amide connected to the core scaffold seem to have overall better pIC_{50} values compared to compounds that have only one of these moieties attached to the core.*”

Docking poses of the 10 compounds that were characterized with higher confidence (both binding and non-binding molecules) are included in the Supplementary Data to allow the reader to compare poses and interactions for themselves, if required.

As suggested, we also performed an MD simulation with one of the compounds that did not bind to or show an effect on FZD₇, in specific compound C476. Data is shown in the new Supplementary Figures S22 and S23. An additional paragraph describing some observations was included in the results section: “To confirm that the disruption of the conformation of the microswitch residues W354^{3x43} and Y478^{6x40} by C407 is not based on a simulation artifact, we additionally ran MD simulations with one of the compounds that was identified to not bind to FZD₇ or affect induced signalling, compound C476. Interestingly, within the first nanoseconds of the simulation C476 shows a similar movement deeper into the 7TMD of FZD₇ as observed for C407, although it cannot form the same polar interactions with S351^{3x40}. This might indicate a generally more favourable binding location for the compounds deep within the 7TMD. However, C476 does not disrupt the conformation of the microswitch residues W354^{3x43} and Y478^{6x40}. On the contrary, the conformation of these residues seems to be more stabilised compared to the apo active simulations and more comparable to the inactive conformation MD simulations, as can also be seen from the RMSF values. These results point further towards the proposed mechanism of action of C407.”

Furthermore, the observations were taken up in the discussion section in context of an SAR: “Structural differences between compounds with and without effect as well as the data from the MD simulations with C407 and non-binder C476 suggest, that a polar moiety, such as an amide, which can form polar interactions with S351^{3x40} might be beneficial for an effective compound. However, it will require more tested compounds and further investigations to understand how to improve affinity and potency.”

2. For rigor, it would be preferable to first perform MD refinement of the C45–FZD7 complex, then use representative structures for subsequent focused screening. This may improve the identification of higher-affinity analogs.

This is indeed a valid suggestion. We agree that it might be beneficial to use MD refined representative structures for a new screening or for a focused screening based on this study to improve the identification of hits and higher affinity analogues. In the presented research we have, however, performed subsequent screenings before deciding to investigate compound dynamics (resulting in hits that were at least improved regarding toxicity) and, hence, this is how we present it here. Additional screenings with MD refined structures are unfortunately out of scope for this manuscript but will be subject of future investigations.

3. While in vitro and biosensor data support C407's activity, its efficacy in disease models (e.g. intestinal tumor organoids or xenografts) is not addressed.

This point is an interesting one. The reviewer appreciated in the intro that our current work is indeed a breakthrough and proof-of-principle because no other validated FZD targeting drugs/NAMs have been established. We have been thinking of the intestinal organoid model, where the addition of high concentrations of C407 should be growth inhibiting or killing the organoid given the possible involvement and requirement of FZD₇. Nevertheless, we have no meaningful pharmacological option to validate that the C407 effect is indeed mediated by FZD₇ (FZD₇ KO would prevent organoid formation, and we cannot compete out C407 with a ligand without efficacy since such a ligand does not exist). We would indeed like to provide this data set, but in vivo/disease models are at this point of the development still too complex to draw rigorous conclusions.

4. C407 shows similar activity across FZD4/5/7. The lack of paralog selectivity should be discussed in terms of therapeutic implications, and structural/sequence comparisons or mutagenesis studies could be conducted to explore the feasibility of achieving selectivity.

C407 is not suitable for a therapeutic approach, and the reviewer is correct in the assessment that paralog-selectivity would be required – or at least homology cluster selectivity. This work presents a breakthrough in itself defining a validated scaffold as NAM acting on FZDs. We feel that the development of paralog selectivity is indeed desirable – although difficult to achieve - but extends the scope of this work.

To explore the feasibility of a FZD paralog selective compound further, **we have added a sequence comparison of any residues within 5Å distance of the potential binding poses of C407 (from docking calculations as well as MD simulations) in Supplementary Table S1 and highlighted the conservation of these residues on the FZD₇ structure in Supplementary Figure S9. Furthermore, we added the following sentences to the results section: “*This is in line with the high degree of evolutionary conservation of Y^{6x51} among FZDs as well as the high overall conservation of residues within the potential binding site*”; and to the discussion: “*Furthermore, it would be of interest to achieve FZD paralog selectivity, although this might be hindered by the overall sequence conservation of FZDs in the region of the potential binding site. Here, it generally seems more likely to achieve selectivity for a subgroup of FZDs than a specific FZD paralog*”.**

5. While TOPFlash data suggest FZD7 as the primary target, the possibility of C407 interacting with other WNT/β-catenin pathway proteins is not excluded.

The most proximate initiator of WNT/ β -catenin signaling apart from WNT receptor activation is the overexpression of DVL, which initiates WNT/ β -catenin in an agonist (WNT)-independent manner. We have performed experiments along this line in combination with C407 at concentrations that inhibit agonist- and FZD-mediated WNT/ β -catenin signaling (TOPflash). Since C407 does not affect the DVL overexpression-induced TOPflash signal, we concluded that other WNT/ β -catenin pathway proteins downstream of DVL are not targeted. This graph was already included in the original manuscript as Fig. S5C:

In addition to these experiments, we have also evaluated the effect of C407 on a TOPflash response induced by overexpression of LRP6 (in absence of FZDs; Δ FZD₁₋₁₀- KO cells) as an orthogonal control (see Figure below). This yielded similar results to the DVL overexpression, that C407 does not reduce the LRP6-induced TOPflash response in absence of FZDs and can, hence, be assumed to act directly on FZDs. These results were omitted in the manuscript to maintain clarity.

6. The proposed ligand entry route via the extracellular side into the 7TMD core warrants more detailed MD-based pathway sampling or adaptive biasing simulations.

We appreciate this suggestion and agree that a more detailed analysis of the ligand entry pathway could yield significant insights. However, at this point there are still too many unknown factors when studying FZDs to investigate this further. For example, the flexibility of the CRD and linker domain might contribute to occlusion of the ligand entry route and thereby impact ligand binding to the core. However, at the current point we have barely any knowledge about CRD positioning and dynamics and, hence, cannot judge or describe this impact further. Therefore, we are convinced that results of such an investigation would be pure speculation and would, furthermore, be difficult to support by wet-lab data.

To reflect these difficulties in describing a ligand binding pathway in the manuscript, **we have added the following sentence in the discussion: “*This suggested binding mechanism is also in line with previous observations that suggest a dynamic opening and closing of this cavity to the extracellular side (Bous et al 2024). However, further investigations to confirm this would be required, considering, for example the flexibility of the CRD impacting the ligand entry due to occlusion effects.*”**

7. Since apo-FZD7 was proposed to stabilize microswitches (e.g., W354^{3x43}, Y478^{6x40}), it is recommended to include comparative MD simulations starting from the 9EPO apo structure to directly quantify differences in residue rigidity and interaction networks. In addition, the author claimed that ligand-bound cryo-EM conformations are more rigid than apo models could be substantiated by providing RMSF values for key residues.

We appreciate this comment and implemented additional analyses. In pursuit of sustainable research, we decided to not run additional MD simulations of 9EPO but use available simulation data of this FZD₇ structure instead (Bous et al. 2024). Although these MD simulations were run using the Charmm36m force field, the comparability of the data with the MD simulations in this manuscript should be given within the small frame we inspect here. Movements of the two microswitch residues were inspected in the same way as done for the other MD simulations. Indeed, as claimed in the manuscript, the two residues were more stabilized in the inactive conformation compared to the active conformation. **The respective data was added in a new Supplementary Figure S21.**

Furthermore, as suggested, we also calculated the RMSF values of Cα atoms and side-chain heavy atoms of the two residues W^{3x43} and Y^{6x40} for all MD simulations. The values point to a similar observation, with smaller movement of the residues in the inactive conformation MDs. Interestingly, the trend was different for non-binder C476, as discussed already above. **The data is included in the new Supplementary Table S4.**

Some additional sentences were included in the results section to reference to the newly included data: “Comparison of the results of an MD simulation of an apo

inactive conformation as performed by Bous et al. (Bous 2024) reveals similar little movement of these residues, while more movement can be observed in the case of the apo active conformation. The rigidity of these two residues in inactive conformations of FZD₇, compared to active conformations is also supported by the lower values of the root-mean-square fluctuation (RMSF) of the side-chain heavy atoms of both residues.”

8. The final C407 binding poses originating from the docking and cryo-EM conformations diverge by 4–6 Å. The manuscript should better explain the observed instability or shift of the cryo-EM pose and how receptor conformation modulates ligand binding.

We thank the reviewer for pointing out the additional need for clarification here. To address this, **we have added some additional sentences in the discussion section:** *“The differences in the observed final poses can likely be attributed to the different starting conformations of the receptor in each case, i.e. an active conformation compared to an inactive conformation, which impacts side chain orientations and flexibility of the receptor. Similarly, the divergent location of the density potentially corresponding to C407 in the cryo-EM data can be attributed to a receptor conformation that is in an inactive state and especially stabilised as a non-natural homodimer. We assume, that the movement of the compound within the receptor is hindered by the rigidity of the inactive, antiparallel dimer, which corresponds to the observations from the MD simulations and the cryo-EM data.”.*

Reviewer #2 (Remarks to the Author)

In this study, Dr. Scharf employed molecular docking to identify small molecules targeting the transmembrane core domain of FZD₇, followed by validation using functional assays, site-directed mutagenesis, molecular dynamics simulations, and cryo-EM analysis. The seeking of small-molecule binders for FZDs is novel, given the current lack of validated ligands for this receptor family, which has significantly impeded the development of FZD-targeted anti-cancer therapies. However, several concerns need to be addressed before the manuscript can be considered for publication.

We thank the reviewer for their summary and appreciation of the significance of the presented study.

1. The density corresponding to the C407 ligand appears weak. The authors should apply

cryo-EM single particle analysis techniques, such as focused classification or local refinement etc., to better resolve and confirm the presence of C407 within the receptor structure.

The reviewer raises a valid point regarding further investigation into the weak density. Indeed, the following procedure yielded what was presented in the original submission. Focused classification was performed using a smaller mask and is detailed in an additional figure below. A loose mask was generated in Relion with a soft edge and imported into an initial 3D classification job (2 classes with Blush regularization, T=4). This resulted in an initial classification comprising 37,796 particles (2.72Å) and 322,926 particles (2.59Å). The second class (322,926 ptcls) was launched into multiple rounds of classification in both Relion and CryoSPARC to compare. The best classes are shown which do not appear to improve the weak density that was initially observed. This could be due to general heterogeneity with the ligand adopting multiple conformations. The conformations appear to be continuous rather than discrete given increasing the regularization parameter (T) did not differentiate two distinct particle subsets. A high T parameter favors the prior or requires a stronger detectable difference to be split into separate classes. An alternative explanation could be that the smaller mask is limiting alignment of the particles which can cut off relevant signal causing noise to dominate. Subsequently a focused mask of the dimer was used to help with alignment and further classifications.

To further clarify the details of 3D classification and local refinement procedure, the following panel was introduced to the SI as a new Fig. S11:

2. It would be valuable to assess the specificity of C407 across other FZD subtypes, such as FZD6, FZD2, and FZD4.

We do not expect C407 to be paralog selective and – even though we have not tested all FZDs yet – we predict that C407 will address all FZD paralogs (as is also suggested by the newly added sequence alignment of the potential binding site region). In the original manuscript we have validated the C407 effect on representative FZD paralogs for FZD homology clusters of FZDs that mediate WNT/ β -catenin signaling (clusters with tested paralog in bold: FZD_{1,2,7}; FZD_{3,6}; FZD_{4,9,10}; FZD_{5,8} – FZD₃ and FZD₆ do not mediate WNT/ β -catenin signaling). Thus, the data for FZD₄ and FZD₅ - as requested by the reviewer - were already presented in the original manuscript in the original Supplementary Figure S8B, C. **Given the reviewer’s request, we performed experiments using the FZD-DEP-Clamp sensor for FZD₂. The measurement for FZD₂ is now added to Supplementary Figure S8 and the results section. We also clarified distinctly, why the paralogs were selected for the experiments. The following sentence was added in the results section: “For this, we selected representative members of each of the homology clusters of FZD paralogs that are involved in WNT/ β -catenin signalling, in specific FZD₂ (FZD_{1,2,7} cluster), FZD₄ (FZD_{4,9,10} cluster), and FZD₅ (FZD_{5,8} cluster).”.**

The additional FZD₂ data included in the supplementary information are shown here:

3. The authors propose that C407 functions as a negative allosteric modulator (NAM) of FZD7 by inhibiting DVL recruitment. It would be informative to compare the structure of C407-bound FZD7 with the recently published FZD4–DEP complex, which may provide mechanistic insights into how C407 modulates receptor activity.

The experiments actually do not argue for C407-mediated interference with DVL recruitment but rather with the disturbance of WNT-induced conformational rearrangements in the core of FZDs that are assessed with the FZD DEP Clamp sensor. The published FZD₄-DEP structure presents most likely the state and conformation of

basally active FZD coupling to DEP. Currently, we do not know how the conformation of the WNT-activated FZD-DEP complex looks like that the FZD DEP Clamp sensors reports on. Thus, we think that the comparison of C407-FZD₇ complex with the FZD₄-DEP complex is not very informative and we hope that the reviewer agrees with us.

Reviewer #3 (Remarks to the Author)

The study by Scharf et al. reports the discovery of C407, the first-in-class negative allosteric modulator of FZD7 that acts by binding to its 7TM domain. This is a rigorous computational, structural, and molecular pharmacology study employing compound docking, in silico library screening, pharmacological assays with BRET- and luminescence-complementation-based conformational and signaling biosensors, site-directed mutagenesis, cryo-EM structure determination, and MD simulations. The authors demonstrate great attention to details, possible nonspecific mechanisms of action (e.g. luciferase inhibition), and alternative interpretations of low-resolution data (e.g. cryo-EM densities).

Unfortunately the presented molecule, C407, is weak and non-selective, and limited efforts of optimizing it via SAR-by-catalog did not lead to more potent NAMs. Nevertheless, this molecule can serve as a proof-of-principle for the proposed model of modulation of receptors in the FZD family, complementing the previously characterized inhibitors of SMO that act through the same mechanism.

The authors did an excellent job addressing potential caveats of the study and employing a combination of experimental and computational approaches to elucidate the mechanism of action of C407. I only have two minor comments.

We thank the reviewer for their positive feedback and the excellent summary of the key points of this study.

1. The authors made compound-bound structures and models available for review, which is commendable. Encouragingly, they also deposited the MD trajectories to <https://gpcrmd.org/>; however, no access code was provided to the reviewers. I encourage the authors to include this info with the final published paper.

We apologise that this information did not reach the reviewers. Deposited MD trajectories (IDs 2339 (docking pose), 2340 (apo-active), 2341 (structure pose) and newly added 2364 (C476)) can currently be accessed using the link <https://gpcrmd.org/dynadb/publications/1629/> and the password “Frizzled7”. Upon

acceptance of the paper, the deposited trajectories will be made openly available, and no password will be required any longer.

2. recommend adding a schematic of the BODIPY-Cyc competition binding assay to Fig 1 and a schematic of the TOPFlash to Fig 2, similar to that of FZD-DEP Clamp biosensor in Fig 2C.

To provide more clarity as with the other assays, we have designed and added schematics of the BODIPY-cyclopamine competition binding assay to Fig. 1 and the TOPFlash assay to Fig. 2. The BODIPY-cyclopamine structure was designed with ChemDoodle freeware with the BODIPY moiety depicted in green.

[editorial note: figure redacted due to third party material]